UPDATE ARTICLE

# The dorsal fan-shaped body is a neurochemically heterogeneous sleep-regulating center in *Drosophila*

**Joseph D. Jones‡, Brandon L. Holder‡, Andrew C. Montgomery, Chloe V. McAdams, Emily He, Anna E. Burns, Kiran R. Eiken, Alex Vogt, Adriana I. Velarde, Alexandra J. Elder, Jennifer A. McEllin, Stephane Dissel** ⊙ *

Division of Biological and Biomedical Systems, School of Science and Engineering, University of Missouri-Kansas City, Kansas City, Missouri, United States of America

‡ These authors share first authorship on this work.
* dissels@umkc.edu

## Abstract

Sleep is a behavior that is conserved throughout the animal kingdom. Yet, despite extensive studies in humans and animal models, the exact function or functions of sleep remain(s) unknown. A complicating factor in trying to elucidate the function of sleep is the complexity and multiplicity of neuronal circuits that are involved in sleep regulation. It is conceivable that distinct sleep-regulating circuits are only involved in specific aspects of sleep and may underlie different sleep functions. Thus, it would be beneficial to assess the contribution of individual circuits in sleep's putative functions. The intricacy of the mammalian brain makes this task extremely difficult. However, the fruit fly *Drosophila melanogaster,* with its simpler brain organization, available connectomics, and unparalleled genetics, offers the opportunity to interrogate individual sleep-regulating centers. In *Drosophila*, neurons projecting to the dorsal fan-shaped body (dFB) have been proposed to be key regulators of sleep, particularly sleep homeostasis. We recently demonstrated that the most widely used genetic tool to manipulate dFB neurons, the 23E10-GAL4 driver, expresses in 2 sleep-regulating neurons (VNC-SP neurons) located in the ventral nerve cord (VNC), the fly analog of the vertebrate spinal cord. Since most data supporting a role for the dFB in sleep regulation have been obtained using 23E10-GAL4, it is unclear whether the sleep phenotypes reported in these studies are caused by dFB neurons or VNC-SP cells. A recent publication replicated our finding that 23E10-GAL4 contains sleep-promoting neurons in the VNC. However, it also proposed that the dFB is not involved in sleep regulation at all, but this suggestion was made using genetic tools that are not dFB-specific and a very mild sleep deprivation protocol. In this study, using a newly created dFB-specific genetic driver line, we demonstrate that optogenetic activation of the majority of 23E10-GAL4 dFB neurons promotes sleep and that these neurons are involved in sleep homeostasis. We also show that dFB neurons require stronger stimulation than VNC-SP cells to promote sleep. In addition, we demonstrate that dFB-induced sleep can consolidate short-term memory (STM) into long-term memory (LTM), suggesting that the benefit of sleep on memory is not circuit-specific. Finally, we show that dFB neurons are neurochemically

**Data availability statement:** All data are available in the main text or the Supporting information apart from Movies 1-16 which are deposited at: https://osf.io/64zh9/?view_only=d058f62c75254556a22f11bd28979287

**Funding:** National Institute of Health 1R01NS130195-01A1 to SD. The funder play no role in the study design, data collection and analysis, decision to publish, or preparation of the manuscript.

**Competing interests:** The authors have declared that no competing interests exist.

**Abbreviations:** AD, activation domain; ChAT, choline acetyltransferase; CI, courtship index; DBD, DNA-binding domain; dFB, dorsal fan-shaped body; LTM, long-term memory; MCFO, MultiColor FlpOut; SD, sleep deprivation; SI, suppression index; SNAP, sleep-nullifying apparatus; STM, short-term memory; TMP, trimethoprim; VAChT, vesicular acetylcholine transporter; vFB, ventral fan-shaped body; VGlut, vesicular glutamate transporter; VLPO, ventrolateral preoptic nucleus; VNC, ventral nerve cord

heterogeneous and can be divided in 3 populations. Most dFB neurons express both glutamate and acetylcholine, while a minority of cells expresses only one of these 2 neurotransmitters. Importantly, dFB neurons do not express GABA, as previously suggested. Using neurotransmitter-specific dFB tools, our data also points at cholinergic dFB neurons as particularly potent at regulating sleep and sleep homeostasis.

## Introduction

Understanding the neural basis of behavior is a major aspect of neurobiology. However, unequivocally assigning a behavior to a specific neuron or group of neurons is not a trivial task. The extreme complexity, diversity, and connectivity of the mammalian nervous system renders this task even more daunting. To study the neural basis of a behavior, an investigator must be able to specifically manipulate a distinct group of cells and monitor the behavior of interest. Such approaches require precise genetic tools that allow for the manipulation of discrete neurons or groups of neurons. While such tools exist in mammalian systems [1], they may not be available for all the diverse types of cells that underlie a specific behavior. It is therefore not surprising that animal models have been extensively used to untangle the neural basis of many different complex behaviors [2]. One such model is the fruit fly *Drosophila melanogaster*, in which multiple binary systems have been developed to access and manipulate specific groups of neurons within the fly nervous system: GAL4/UAS [3], LexA/LexAop [4], and QF/QUAS [5]. Of these 3 systems, the GAL4/UAS system has been by far the most extensively used. However, the expression patterns of GAL4 drivers are often not restricted enough to clearly link a behavior to specific neurons. In such cases, refinement of GAL4 expression pattern can be achieved by employing the intersectional Split-GAL4 technology [6].

Sleep is a behavior that has been observed in a multitude of species ranging from jellyfish to humans [7]. A priori, sleep could appear to be a detrimental activity as it competes with other motivated behaviors, such as feeding, mating, or parenting, and renders organisms defenseless against potential predators. Despite these negative outcomes, sleep has been maintained throughout evolution, emphasizing its essential value [8]. Sleep is regulated by 2 processes, the circadian clock which gates the occurrence of sleep and the sleep homeostat which controls the intensity and duration of sleep in response to prior wakefulness [9]. Since the first characterization of sleep in *Drosophila* [10,11], multiple studies have emphasized a high level of conservation of sleep mechanisms and regulation between flies and mammals [12]. Like the mammalian system, sleep-regulating centers are found in many areas in the *Drosophila* brain [12,13]. Among them, neurons that project to the dorsal fan-shaped body (dFB) have attracted a lot of attention. Previous studies demonstrated that increasing the activity of neurons contained in the C5-GAL4, 104y-GAL4, and C205-GAL4 drivers strongly promotes sleep [14,15]. While the expression patterns of these 3 independent drivers are broad, they show prominent overlap in the dFB [14]. Based on these observations, the authors concluded that it is likely that the dFB plays a role in regulating sleep but could not rule out a role for neurons outside the dFB [14]. In addition, further studies established that reducing the excitability of 104y-GAL4 neurons decreases sleep [16,17]. Highlighting the strong interaction between sleep and memory, activation of 104y-GAL4 neurons consolidates short-term memory (STM) to long-term memory (LTM) [14,18] and restores STM in the classical memory mutants *rutabaga* and *dunce* [19]. Further work supported a role for the dFB in sleep homeostasis by demonstrating that sleep deprivation increases the excitability of 104y-GAL4 dFB neurons [16]. Taken together, these data pointed at 104y-GAL4 expressing neurons, most likely dFB cells, as important modulators of sleep. However, it was demonstrated that 104y-GAL4 also expresses

in neurons projecting to the ventral fan-shaped body (vFB) and that these neurons modulate sleep and are involved in the sleep-dependent consolidation of STM into LTM [18], making the conclusions about the role of 104y-GAL4 dFB neurons uncertain. More recent studies using 23E10-GAL4, a more restrictive driver to manipulate and monitor dFB neurons [20,21], proposed that increasing sleep pressure switches dFB neurons from an electrically silent to an electrically active state, and that this process is regulated by dopaminergic signaling to the dFB [20] and the accumulation of mitochondrial reactive oxygen species in dFB neurons [22]. Because of their physiological properties, dFB neurons have been proposed to be the fly functional analog to the ventrolateral preoptic nucleus (VLPO), a key center regulating sleep homeostasis in the mammalian brain [16,23].

Chronic and acute activation of 23E10-GAL4 neurons increases sleep [18,24–28]. Since it has a relatively restricted expression pattern and a strong capacity to modulate sleep, 23E10-GAL4 (as it relates to sleep) is seen as a dFB-specific driver by most in the scientific community. However, our recent work, aimed at identifying individual sleep-regulating 23E10-GAL4 expressing neurons, demonstrated that this driver is not as dFB-specific as previously believed. In particular, we demonstrated that 23E10-GAL4 expresses in 2 cholinergic sleep-promoting cells (VNC-SP neurons) located in the ventral nerve cord (VNC) [27]. Additional work from another laboratory confirmed that there are VNC-localized sleep-promoting neurons in the 23E10-GAL4 driver [28]. Together, these findings have raised some serious questions about the role, if any, of the dFB in sleep regulation, especially if these data were obtained using 23E10-GAL4. In fact, 2 recent studies suggest that the dFB has no sleep-regulating capacity [28,29]. However, for one of these studies, the conclusions made by the authors are based on data obtained with non-dFB-specific tools and a very mild sleep deprivation protocol [28]. The second study employed a very brief and mild optogenetic activation of all 23E10-GAL4 neurons [29]. In the current study, we sought to assess the role of the dFB in sleep regulation by conducting a targeted, intersectional Split-GAL4 screen [6] focused on 23E10-GAL4 dFB neurons. We report here that sleep-promoting VNC-SP neurons are present in most Split-GAL4 lines we created, making it difficult to assess the role of the dFB using these lines. However, we identified a dFB-specific Split-GAL4 line that can be used to manipulate most 23E10-GAL4 dFB neurons. Using this novel driver, we demonstrate that 23E10-GAL4 dFB cells regulate sleep and play a role in sleep homeostasis. Importantly, dFB neurons require relatively strong activation protocols to promote sleep, stronger than what we found for VNC-SP neurons [27]. We also show that dFB-induced sleep promotes the consolidation of STM to LTM. Additionally, our work reveals that dFB neurons are neurochemically heterogeneous and that cholinergic dFB neurons play an important role in sleep regulation.

## Results

### A dFB-based Split-GAL4 screen identifies multiple sleep-promoting lines

To assess the role of 23E10-GAL4 dFB neurons in sleep regulation, we designed a Split-GAL4 screen [6] focused on the dFB. The Split-GAL4 technology separates the functional GAL4 transcription factor into 2 non-functional fragments, a GAL4 DNA-binding domain (DBD) and an activation domain (AD). Expression of the AD and DBD fragments is controlled by different enhancers and the functional GAL4 transcription factor is reconstituted only in the cells in which both fragments are expressed [6]. We obtained 20 different AD lines based on their associated GAL4 line's expression in the dFB, as observed using the Janelia FlyLight website [21,30]. Since the goal of our screen was to identify the contribution of 23E10-GAL4 dFB neurons in sleep regulation, we designed a targeted approach by combining these individual AD lines to a 23E10-DBD line [30,31], thus creating 20 new Split-GAL4 lines named

FBS for F̲an-S̲haped B̲ody S̲plits (see S1 Table for a description of these lines). These newly created lines were screened behaviorally for their ability to modulate sleep and anatomically to identify their expression patterns. To activate the cells, each individual Split-GAL4 FBS line was crossed to: (1) a line expressing both the thermogenetic TrpA1 cation channel [32] and an mCD8GFP construct; and (2) the optogenetic CsChrimson cation channel [33].

For the thermogenetic screen, flies were maintained at 22 °C for 2 days before raising the temperature to 31 °C at the beginning of day 3 for a duration of 24 h. Temperature was then lowered to 22 °C on day 4 for recovery (S1A Fig). For each individual fly, we calculated the percentage of total sleep change between activation day (day 3) and baseline day (day 2) (S1A Fig). As a control for this screen, we used an enhancerless AD construct (created in the vector used to make all the AD lines) combined with 23E10-DBD, since it is the common element in all the Split-GAL4 lines analyzed in this screen. As seen in S1B Fig, acute thermogenetic activation of the neurons contained in 4 of the 20 FBS lines led to significant increases in sleep in female flies (FBS42, FBS45, FBS53, and FBS68), when compared with controls. Individual sleep traces for these 4 lines are shown in S1D–S1H Fig. Sleep profiles of the other 16 FBS lines are shown in S2 Fig. Importantly, analysis of activity counts during awake time reveals that increases in sleep are not caused by a reduction in locomotor activity (S3A Fig), ruling out motor deficits or paralysis. On the contrary, 2 of the sleep-inducing lines even showed an increase in waking activity upon neuronal activation. While increases in total sleep are indicative of increased sleep quantity, this measurement does not provide information about sleep quality or sleep depth. Previous work proposed that increased sleep bout duration is an indication of increased sleep depth [34]. To assess whether sleep quality is modulated when activating FBS lines, we analyzed sleep consolidation during the day and night in these flies. As seen in S3B Fig, daytime sleep bout duration is significantly increased upon thermogenetic activation in 7 out of 20 FBS lines (FBS28, FBS42, FBS45, FBS53, FBS68, FBS72, and FBS84). Interestingly, 3 of those 7 lines did not show an increase in total sleep (FBS28, FBS72, and FBS84). During the nighttime, the effect of high temperature on sleep is obvious as most lines, including the control, display a significant decrease in sleep bout duration when raising the temperature to 31 °C (S3C Fig). These data agree with previous studies documenting that changes in temperature modulate sleep architecture and that high temperature disturbs sleep at night [35–38]. Only 4 of the 20 FBS lines (FBS42, FBS45, FBS53, and FBS68) maintain similar nighttime sleep bout durations before and during thermogenetic activation. Importantly, in 3 of these 4 lines, nighttime sleep bout duration is significantly increased at 31 °C when compared to control flies (S3C Fig).

Since sleep in *Drosophila* is sexually dimorphic [39–41], we systematically assessed male flies in our experiments. As seen in S4A Fig, thermogenetic activation increases total sleep in males in 7 out of the 20 FBS lines, including the 4 lines that increase total sleep in females (FBS42, FBS45, FBS53, FBS68, FBS72, FBS81, and FBS84). These effects on total sleep are not caused by motor deficits or paralysis as waking activity is not reduced in long sleeping male flies (S4B Fig). Daytime sleep bout duration is significantly increased in 6 out of these 7 lines (S4C Fig). When examining nighttime sleep, thermogenetic activation of FBS45 and FBS68 significantly increases bout duration while FBS42, FBS53, FBS81, and FBS84 show no difference between 22 and 31 °C (S4D Fig). Altogether, our thermogenetic approach identified 8 FBS lines that increase total sleep and/or sleep bout duration when activated (S2 Table).

Because of the strong effect that temperature has on sleep, we sought to confirm our thermogenetic findings using an alternative method to manipulate FBS lines. The logic behind our thinking is that it may be difficult to fully describe and characterize the sleep behaviors of our FBS lines when temperature itself has such a profound effect on sleep. Furthermore, a recent study demonstrated that some of the effects of high temperature on sleep are mediated

by GABAergic transmission on dFB neurons [38]. We thus undertook an optogenetic screen using CsChrimson [33]. Our optogenetic experimental setup is described in S1A Fig. Notably, for neurons to be activated when expressing CsChrimson, flies need to be fed all trans-retinal and the neurons must be stimulated with a 627 nm LED light source [33]. Thus, optogenetic approaches provide a much better control on the timing and parameters of activation. Our regular optogenetic activation screen (consisting of a pulse cycle of [5 ms on, 95 ms off] × 20 with a 4 s delay between pulse cycles) gave results that are mostly identical to the thermogenetic approach. That is, all 4 sleep-promoting lines identified using TrpA1 also increase total sleep when activated with CsChrimson in retinal-fed female flies (FBS42, FBS45, FBS53, and FBS68) (S1C Fig). Interestingly, optogenetic activation of neurons contained in FBS70 and FBS72 is sleep-promoting in retinal-fed female flies while it was not using TrpA1 (S1B and S1C Fig). Sleep profiles of all FBS lines subjected to optogenetic activation are shown in S5 Fig. Looking at males, optogenetic activation increases total sleep in all 7 sleep-promoting lines identified using TrpA1 (FBS42, FBS45, FBS53, FBS68, FBS72, FBS81, and FBS84), as well as an 8th line (FBS33) (S6A Fig). Importantly, these optogenetic sleep-promoting effects are not the result of locomotor deficits (S6B Fig for males and S7A Fig for females).

Examination of sleep bout duration in retinal-fed flies indicates that most sleep-promoting FBS lines increase sleep consolidation during the day (S6C Fig for males and S7B Fig for females) and during the night (S6D Fig for males and S7C Fig for females). Interestingly, some FBS lines that did not increase total sleep upon optogenetic activation increased daytime sleep bout duration in females (FBS57, FBS60, FBS81, and FBS84) (S7B Fig) and in males (FBS35, FBS64, and FBS87) (S6C Fig). In addition, optogenetic activation of neurons contained within FBS58 increases nighttime sleep bout duration in females (S7C Fig). These data suggest that these lines may also express in sleep-regulating neurons. Importantly, analysis of sleep parameters in vehicle-fed flies demonstrate that the sleep phenotypes we observed are specific to retinal-fed and LED stimulated flies (S8 Fig for females and S9 Fig for males).

A summary of sleep-modulating effects for all FBS lines in both activation protocols is provided in S2 Table. Taken together, our thermogenetic and optogenetic screens revealed that activating neurons contained within 16 FBS lines increases at least 1 sleep parameter. Among these 16 lines, 4 are consistently increasing total sleep and sleep consolidation in females (FBS42, FBS45, FBS53, and FBS68) and 7 in males (FBS42, FBS45, FBS53, FBS68, FBS72, FBS81, and FBS84), using both activation protocols (S2 Table).

## VNC-SP neurons are present in most FBS lines

Having identified 16 different Split-GAL4 lines that significantly increase at least 1 sleep parameter when thermogenetically or optogenetically activated, we sought to identify the neurons that are contained within these lines. As seen in S1D–S1H and S2 Figs, 19 out of 20 lines express in dFB neurons, with only FBS25 showing no expression at all in the brain and VNC. Since FBS25 expresses in no neurons at all, it is not surprising that no sleep changes were seen using both activation protocols, as FBS25 should behave like a control. The number of dFB neurons contained within different FBS lines ranges from 4 to 27 per brain on average (S1 Table). Most lines show very little additional expression in the brain and there is no consistency in these non-dFB labeled cells between different lines. However, 18 out of the 19 lines that express in dFB neurons also express in cells in the metathoracic ganglion of the VNC. This is not surprising as our previous study showed that the 23E10-GAL4 driver, on which this Split-GAL4 screen is based, expresses in 4 neurons located in the metathoracic ganglion of the VNC, consisting of 2 TPN1 neurons [42] and the 2 VNC-SP cells [27]. Most FBS lines (17 out of 20) express in neurons that have anatomical features similar to VNC-SP neurons in the VNC and in the brain, including very typical processes that we previously

named "bowtie" [27]. These "bowtie" processes are located extremely close to the axonal projections of dFB neurons, making their visualization sometimes difficult depending on the strength of dFB projections' staining. Importantly, all 16 FBS lines that increase at least 1 sleep parameter when activated express in these VNC-SP-like cells, in addition to dFB neurons. Furthermore, most FBS lines (14 out of 20) also express in cells that appear to be TPN1 neurons [42].

Our previous study demonstrated that VNC-SP neurons are cholinergic and that the expression of the Split-GAL4 repressor KZip⁺ [43] under the control of a ChAT-LexA driver effectively blocks the accumulation of reconstituted GAL4 in VNC-SP neurons [27]. Since we observed neurons reminiscent of VNC-SP in all 4 FBS lines that increase sleep when thermogenetically activated in females, we wondered whether VNC-SP neurons are the cells responsible for the sleep increase in these lines. To address this possibility, we expressed TrpA1 and GFP in FBS42, FBS45, FBS53, and FBS68 neurons while simultaneously driving the expression of the KZip⁺ repressor in ChAT-LexA cells. As seen in S10A–S10D Fig, expressing the KZip⁺ repressor in cholinergic cells abolishes GFP expression in the metathoracic ganglion in all *FBS>UAS-GFP; ChAT-LexA>LexAop2-KZip⁺* flies. These data demonstrate that VNC-SP neurons are present in the expression pattern of all 4 FBS lines tested. When assessing sleep, expressing the KZip⁺ repressor in ChAT-LexA cells prevents the increase in sleep observed in all *FBS>UAS-TrpA1* flies tested (S10E Fig). These data suggest that the neurons responsible for the sleep increase observed when thermogenetically activating FBS42, FBS45, FBS53, and FBS68 cells are the cholinergic VNC-SP neurons. However, since we previously demonstrated that some 23E10-GAL4 dFB neurons are cholinergic [27], it is possible that cholinergic 23E10-GAL4 dFB neurons may also participate in these sleep phenotypes. Based on these data, and our previous work showing that 23E10-GAL4 expresses in VNC-SP cells [27], we believe that all 16 FBS sleep-promoting lines express in VNC-SP neurons, making it impossible for us to assess whether 23E10-GAL4 dFB neurons modulate sleep using these lines.

However, if VNC-SP neurons are present in most FBS lines, why do not we see any sleep changes with one of them (FBS1)? In addition, why are some VNC-SP expressing FBS lines more potent than others at promoting sleep? We hypothesized that these differences may be explained by differences in strength of expression of the reconstituted GAL4 in VNC-SP neurons in FBS lines. To address this, we used a GFP-DD construct which is normally degraded by the proteasome [44]. Degradation is blocked when feeding the flies trimethoprim (TMP) [44] and therefore strength of expression can be assessed. We used 23E10-GAL4 and the VNC-SP Split-GAL4 line previously described [27] as controls to compare expression levels in the VNC-SP neurons of different FBS lines. Flies were maintained on standard food until they reached 5–6 days of age. Next, half of the flies were fed either vehicle control (DMSO) or TMP, to protect GFP-DD from degradation, for 24 h (S11A Fig). Flies were then dissected and GFP levels were measured in the metathoracic ganglion of the VNC. As seen in S11B and S11C Fig, feeding TMP leads to strong increases in GFP signal in VNC-SP neurons when GFP-DD is expressed using 23E10-GAL4, VNC-SP Split, and FBS45. When GFP-DD is expressed in FBS33 cells, we observed a moderate increase in GFP signal intensity. With 2 lines, FBS1 and FBS58, we saw no statistical differences between DMSO-fed and TMP-fed flies. The strength of expression in VNC-SP neurons correlates well with the magnitude of sleep increases seen when activating the different lines (S2 Table). FBS45 is strongly sleep-promoting when thermogenetically and optogenetically activated, while FBS33 only increases total sleep and daytime sleep bout duration in males using optogenetic activation. Finally, FBS58 has a very modest impact on sleep while FBS1 has none. Thus, we conclude that within our FBS lines, the strength of expression in VNC-SP neurons dictates how potently a given line can modulate sleep.

When combining behavioral and anatomical data, we conclude that the numbers of dFB neurons contained within a specific FBS line cannot be predictive of whether the line can strongly promote sleep upon neuronal activation. For example, FBS68 strongly increases all sleep parameters in both sexes and activation protocols while only expressing in 3–6 dFB neurons (S1H Fig and S1 and S2 Tables). Conversely, FBS60 contains 22–27 dFB neurons and only increases daytime sleep bout duration in females using optogenetic activation (S2J Fig and S1 and S2 Tables). The anatomical feature that is predictive of whether an FBS line can modulate sleep or not is the presence/absence of VNC-SP neurons within the pattern of expression. In addition, how strongly a line expresses in VNC-SP neurons dictates how strongly sleep-promoting that line is (S11B and S11C Fig).

In conclusion, our activation screens identified 16 FBS lines, expressing in diverse numbers of dFB neurons, that modulate at least 1 sleep parameter (S2 Table). However, all 16 lines contain the previously identified sleep-promoting VNC-SP neurons. Thus, these lines are not well suited to assess the role of 23E10-GAL4 dFB neurons in sleep.

## Identification of a dFB-specific Split-GAL4 line

In our screen, we only identified 2 FBS lines that express in dFB neurons but not in VNC-SP cells (FBS6 and FBS41). No sleep parameters are increased when activating neurons contained within either line. However, both lines only express in a reduced number of dFB neurons (S2 Fig and S1 Table). Thus, these 2 lines are not fully recapitulating the dFB expression pattern of the 23E10-GAL4 driver. Fortunately, we identified an additional FBS line (84C10-AD; 23E10-DBD which we named dFB-Split) that expresses in 18–27 dFB neurons per brain and no other cells in the brain or VNC (Fig 1A, S1 and S2 Movies, and S1 Table). Since we previously reported that 23E10-GAL4 reliably labels 23–30 dFB neurons in our confocal microscopy experiments [27], we conclude that dFB-Split expresses in 78% to 90% of 23E10-GAL4 dFB neurons. Additionally, dFB-Split does not express in the wings, legs, gut, or ovaries (S12 Fig). Though not being a complete replication of all 23E10-GAL4 dFB neurons, we hypothesized that the dFB-Split line is a good tool to assess a role for 23E10-GAL4 dFB neurons in sleep regulation. For clarity, we will refer to the dFB cells contained in the dFB-Split expression pattern as $dFB^{23E10 \cap 84C10}$ neurons.

## Activation of $dFB^{23E10 \cap 84C10}$ neurons promotes sleep

Having identified a genetic driver that is dFB-specific and includes the majority of the 23E10-GAL4 dFB neurons, we sought to investigate the contribution of these cells to sleep. First, we employed a 1 Hz optogenetic activation protocol and found no effects on sleep in female flies (Fig 1B and 1C). Next, we used our regular optogenetic protocol (consisting of a pulse cycle of [5 ms on, 95 ms off] × 20 with a 4 s delay between pulse cycles) and again obtained no sleep increases when activating $dFB^{23E10 \cap 84C10}$ neurons in females (Fig 1D and 1E). Note that this activation protocol is sufficient to increase sleep using multiple other drivers (S1C Fig) and the VNC-SP Split-GAL4 [27], indicating that it is effective at activating some neurons. Why then does this protocol not cause an effect in $dFB^{23E10 \cap 84C10}$ neurons? Previous studies have demonstrated that the dFB is under strong dopaminergic inhibition [15,17,20,45] and that dopamine is a key factor regulating whether dFB neurons are active or silent [20]. In addition, it was shown that sleep deprivation increases the activity of dFB neurons, switching them to the active state [16]. We hypothesize that in our experiments, sleep pressure is low, as flies are allowed to sleep normally before activation. Thus, the inhibitory activity of dopaminergic inputs to the dFB should be relatively strong and it is possible that our regular activation protocol may not be sufficient to overcome this inhibition. To test this possibility, we decided to

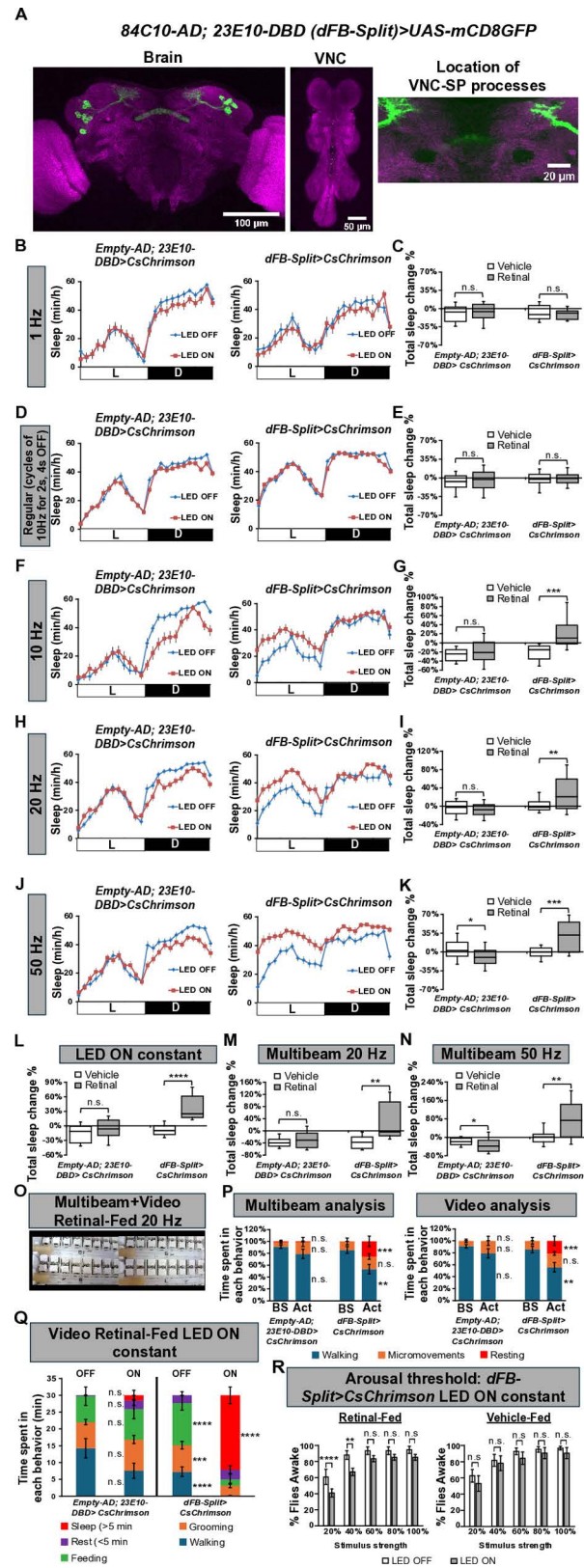

**Fig 1. Activation of dFB**[23E10∩84C10] **neurons promotes sleep. (A)** Representative confocal stack of a female *84C10AD; 23E10-DBD (dFB-Split)>UAS-mCD8GFP* brain (left panel), VNC (middle panel), and the location where VNC-SP "bowtie" processes are seen in many FBS lines, but not in dFB-Split (right panel). We observed 22.52 ± 0.59 (*n* = 23)

dFB neurons in *dFB-Split>UAS-mCD8GFP* brains. Green, anti-GFP; magenta, anti-nc82 (neuropile marker). **(B)** Sleep profile in minutes of sleep per hour for day 2 (LED OFF, blue line) and day 3 (LED ON, red line) for retinal-fed empty control (Empty-AD; 23E10-DBD) and dFB-Split female flies expressing CsChrimson subjected to a 1 Hz optogenetic activation protocol. **(C)** Box plots of total sleep change in % ((total sleep on day 3-total sleep on day 2/ total sleep on day 2) × 100) for female control (Empty-AD; 23E10-DBD) and dFB-Split flies expressing CsChrimson under 1 Hz optogenetic activation (cycles of 5 ms LED ON, 995 ms LED OFF). Two-way ANOVA followed by Sidak's multiple comparisons revealed no differences between control and dFB-Split. n.s. = not significant. *n* = 20-29 flies per genotype and condition. **(D)** Sleep profile in minutes of sleep per hour for day 2 (LED OFF, blue line) and day 3 (LED ON, red line) for retinal-fed empty control (Empty-AD; 23E10-DBD) and dFB-Split female flies expressing CsChrimson subjected to our regular optogenetic activation protocol (5 ms LED ON, 95 ms LED OFF, with a 4 s delay between pulses). **(E)** Box plots of total sleep change in % ((total sleep on day 3-total sleep on day 2/total sleep on day 2) × 100) for female control (Empty-AD; 23E10-DBD) and dFB-Split flies expressing CsChrimson under regular optogenetic activation. Two-way ANOVA followed by Sidak's multiple comparisons revealed no differences between control and dFB-Split. n.s. = not significant. *n* = 59-82 flies per genotype and condition. **(F)** Sleep profile in minutes of sleep per hour for day 2 (LED OFF, blue line) and day 3 (LED ON, red line) for retinal-fed empty control (Empty-AD; 23E10-DBD) and dFB-Split female flies expressing CsChrimson subjected to a 10 Hz optogenetic activation protocol (cycles of 5 ms LED ON, 95 ms LED OFF). **(G)** Box plots of total sleep change in % for female control (Empty-AD; 23E10-DBD) and dFB-Split flies expressing CsChrimson under 10 Hz optogenetic activation. Two-way ANOVA followed by Sidak's multiple comparisons revealed that sleep is significantly increased in *dFB-Split>CsChrimson* females. ****P* < 0.001, n.s. = not significant. *n* = 22-24 flies per genotype and condition. **(H)** Sleep profile in minutes of sleep per hour for day 2 (LED OFF, blue line) and day 3 (LED ON, red line) for retinal-fed empty control (Empty-AD; 23E10-DBD) and dFB-Split female flies expressing CsChrimson subjected to a 20 Hz optogenetic activation protocol (cycles of 5 ms LED ON, 45 ms LED OFF). **(I)** Box plots of total sleep change in % for female control (Empty-AD; 23E10-DBD) and dFB-Split flies expressing CsChrimson under 20 Hz optogenetic activation. Two-way ANOVA followed by Sidak's multiple comparisons revealed that sleep is significantly increased in *dFB-Split>CsChrimson* females. ***P* < 0.01, n.s. = not significant. *n* = 26-38 flies per genotype and condition. **(J)** Sleep profile in minutes of sleep per hour for day 2 (LED OFF, blue line) and day 3 (LED ON, red line) for retinal-fed empty control (Empty-AD; 23E10-DBD) and dFB-Split female flies expressing CsChrimson subjected to a 50 Hz optogenetic activation protocol (cycles of 5 ms LED ON, 15 ms LED OFF). **(K)** Box plots of total sleep change in % for female control (Empty-AD; 23E10-DBD) and dFB-Split flies expressing CsChrimson under 50 Hz optogenetic activation. Two-way ANOVA followed by Sidak's multiple comparisons revealed that sleep is significantly increased in *dFB-Split>CsChrimson* females. **P* < 0.05, ****P* < 0.001. *n* = 25-38 flies per genotype and condition. **(L)** Box plots of total sleep change in % for female control (Empty-AD; 23E10-DBD) and dFB-Split flies expressing CsChrimson under constant optogenetic activation. Two-way ANOVA followed by Sidak's multiple comparisons revealed that sleep is significantly increased in *dFB-Split>CsChrimson* females. n.s. = not significant, *****P* < 0.0001. *n* = 17-27 flies per genotype and condition. **(M)** Box plots of total sleep change in % for female control (Empty-AD; 23E10-DBD) and dFB-Split flies expressing CsChrimson under 20 Hz optogenetic activation obtained with the DAM5H multibeam system. Two-way ANOVA followed by Sidak's multiple comparisons revealed that sleep is significantly increased in *dFB-Split>CsChrimson* females. n.s. = not significant, ***P* < 0.01. *n* = 12-15 flies per genotype and condition. **(N)** Box plots of total sleep change in % for female control (Empty-AD; 23E10-DBD) and dFB-Split flies expressing CsChrimson under 50 Hz optogenetic activation obtained with the DAM5H multibeam system. Two-way ANOVA followed by Sidak's multiple comparisons revealed that sleep is significantly increased in *dFB-Split>CsChrimson* females. **P* < 0.05, ***P* < 0.01. *n* = 35-54 flies per genotype and condition. **(O)** Setup for combined multibeam and video analysis. **(P)** Multibeam (left) and video analysis (right) of retinal-fed control and *dFB-Split>UAS-CsChrimson* female flies. Recordings were performed between ZT3-5 for 10 min of baseline (Bs) followed by 10 min of 20 Hz optogenetic activation (Act). For both multibeam and video analysis, data was analyzed in 1 min bin. Flies could perform the following 3 behaviors, walking, micromovements (in place movements like feeding or grooming) or rest (no movements at all), and percentage of time spent in each behavior over 10 min is shown. If a fly walks during a minute bin, the output for that minute is walking, independently of other behaviors that could be performed during the same minute. If a fly performs a micromovement during a minute bin, but shows no walking, the output for that minute is micromovements. If a fly shows no walking or micromovements for a minute bin, the output for that minute is resting. For video analysis, behaviors were manually scored. Two-way repeated measures ANOVA followed by Sidak's multiple comparisons test found that LED activated retinal-fed *dFB-Split>UAS-CsChrimson* flies spend less time walking and more time resting than on baseline. No changes in micromovements were observed. No changes were seen in control flies. Similar results were obtained with multibeam and video analysis. n.s. = not significant, ** *P* < 0.01, *** *P* < 0.001. *n* = 11-12 flies for each genotype. **(Q)** Video analysis of retinal-fed control and *dFB-Split>UAS-CsChrimson* female flies. Recording was performed at ZT1-2 for 30 min on consecutive days during baseline (OFF) and activation (ON). Behaviors were manually scored, and amount of time spent in each behavior over 30 min is shown. Two-way ANOVA followed by Sidak's multiple comparisons test found that retinal-fed *dFB-Split>UAS-CsChrimson* flies with LED ON sleep significantly more than on baseline day (OFF), **** *P* < 0.0001, but spend less time walking (*****P* < 0.0001), grooming (****P* < 0.01), or feeding (*****P* < 0.0001). Rest (periods of inactivity shorter than 5 min) are not different. We observed no differences in time spent in all behaviors for control flies between activation and baseline days. n.s. = not significant. *n* = 8 flies for each genotype. **(R)** Arousal threshold in vehicle-fed and retinal-fed *dFB-Split>UAS-CsChrimson* female flies. Percentage of flies awakened by a stimulus of

increasing strength (20%, 40%, 60%, 80%, or 100% of maximum strength) with and without 627 nm LEDs stimulation. Two-way ANOVA followed by Sidak's multiple comparisons indicates that activation of dFB$^{23E10\cap84C10}$ neurons reduce the responsiveness to the 20% and 40% stimulus strength when compared with non-activated flies. No difference in responsiveness is seen at the strongest stimulus (60%, 80%, and 100%). Two-way ANOVA followed by Sidak's multiple comparisons indicates that in vehicle-fed flies no difference in responsiveness is seen between LED stimulated and non-stimulated flies. ****$P < 0.0001$, ***$P < 0.001$, n.s. = not significant. $n$ = 16-33 flies per genotype and condition. The raw data underlying parts C, E, G, I, K, L, M, N, P, Q, and R can be found in S1 Data. AD, activation domain; DBD, DNA-binding domain; dFB, dorsal fan-shaped body; VNC, ventral nerve cord.

increase the intensity of the optogenetic activation protocol. As seen in Fig 1F and 1G, a 10 Hz activation of dFB$^{23E10\cap84C10}$ neurons increases total sleep in females. In addition, it significantly increases daytime sleep bout duration (S13B Fig). These effects are not due to a locomotor deficit as waking activity is unaffected by 10 Hz optogenetic activation in females (S13A Fig). Importantly, this increase in daytime sleep consolidation is not seen in vehicle-fed flies (S13E). We found similar sleep-promoting effects when looking at *dFB-Split>CsChrimson* females and males subjected to a 20 Hz optogenetic activation (Fig 1H and 1I for females and S14G Fig for males). In addition, a 20 Hz optogenetic activation increases daytime and nighttime sleep bout duration in female and male flies (S14B and S14C Fig for females and S14I and S14J Fig for males). Again, these effects are not due to a locomotor deficit as waking activity is unaffected by a 20 Hz activation (S14A Fig for females and S14H Fig for males).

We next used a 50 Hz activation protocol. As seen in Fig 1J and 1K, total sleep is increased in *dFB-Split>CsChrimson* females when activated with a 50 Hz protocol. Furthermore, daytime and nighttime sleep bout durations are increased in females using this activation protocol (S15B and S15C Fig). These effects are not the result of abnormal locomotor activity (S15A Fig) and are not seen in vehicle-fed flies (S15E and S15F Fig). Similar behavioral data was obtained when activating dFB$^{23E10\cap84C10}$ neurons with a 50 Hz optogenetic protocol in male flies (S15G–S15M Fig). Since our data suggested a direct relationship between the intensity of the optogenetic activation protocol and the capacity of dFB$^{23E10\cap84C10}$ neurons to increase sleep, we decided to turn the 627 nm LEDs on constantly. As seen in Fig 1L, constant LED activation of dFB$^{23E10\cap84C10}$ neurons significantly increases sleep in females, indicating that once the activation protocol is sufficient (above 10 Hz), dFB$^{23E10\cap84C10}$ neurons can reliably promote sleep.

Recent studies have proposed that activation of all 23E10-GAL4 expressing neurons (including dFB$^{23E10\cap84C10}$ cells) promotes micromovements, such as grooming, rather than sleep [29,46]. It is important to note that these conclusions were made using either a thermogenetic approach that may not be sufficient (29° instead of 31° for TrpA1 activation) or a very brief (5 min) 1 Hz optogenetic activation protocol. As mentioned above, we showed that a 24 h long 1 Hz optogenetic activation of dFB$^{23E10\cap84C10}$ neurons was insufficient to increase sleep (Fig 1B and 1C). To further validate that sufficient optogenetic activation of dFB$^{23E10\cap84C10}$ neurons increases sleep, and not any other behavior (feeding, grooming, or in-place micromovements) that could be falsely registered as sleep by the single beam DAM2 system, we employed the more sensitive multibeam activity monitors (DAM5H, Trikinetics). These monitors contain 15 independent infrared beams, separated by 3 mm across the tube length, resulting in flies being continuously monitored by at least 1 beam during the experiment. First, we sought to validate the sensitivity of the multibeam system to detect different behaviors. We loaded 19 wild-type *Canton-S* flies in 65-mm long glass tubes and monitored behavior for 10 min using the multibeam system and video recording concurrently (S16A Fig). Video analysis revealed that flies always perform one of 7 different behaviors: rest (total immobility), walking (movement), feeding (micromovement), grooming (micromovement), posture change (a small dip of the abdomen, micromovement), proboscis extension (micromovement), and

single leg movement (micromovement) (S16B Fig, Movies 1 and 2 available on https://osf.io/64zh9/?view_only=d058f62c75254556a22f11bd28979287). Our analysis of the multibeam data obtained from the 19 flies revealed that when considering all 7 behaviors, the multibeam sensitivity is 88.9% (S16C Fig). When looking at individual behaviors, sensitivity is above 90% for rest, walking, feeding, and grooming (S16C Fig). However, the multibeam performs poorly for detection of posture change, proboscis extension, and single leg movement (S16C Fig). Notably, these 3 behaviors are extremely subtle with only a very minor movement of one body part, and do not resemble the jerky multiple legs movements that have been described for 1 Hz activation of all 23E10-GAL4 neurons [29]. Additionally, the accuracy of the multibeam system is above 90% for walking and micromovements and 78% for rest (S16C Fig).

Importantly, for walking, the 8.9% of walking events not picked by the multibeam system are scored as micromovements, so they are not falsely registered as rest. Taking this into account, it brings the sensitivity of the multibeam system for walking/micromovements to 91.5%. Missed feeding, grooming, posture change, proboscis extension, and single leg movement events are labeled as rest by the multibeam system.

Interestingly, we observed that posture change, proboscis extension, and single leg movement events are always surrounded by periods of rest (S16D Fig top panels and S16E Fig, bottom panels, Movies 1 and 2 available on https://osf.io/64zh9/?view_only=d058f62c75254556a22f11bd28979287). Previous studies have suggested that such events are associated with sleep [29,47], suggesting that a failure of the multibeam system to detect them may not lead to a dramatic overestimation of sleep, especially considering the low frequency of such behaviors (S16B Fig). If we consider that posture change, proboscis extension, and single leg movement are part of sleep, the accuracy of the multibeam system for correctly detecting rest jumps to 95.9% (S16C Fig).

Finally, for us to falsely label a period of micromovements that has not been associated with sleep (feeding or grooming) as sleep, we would need the multibeam to fail to detect any of these micromovements for 5 consecutive minutes, which is something we have not seen in the 190 min of data we analyzed. In addition, we only detected 1 occurrence where a failure to detect a feeding event would have led to the mislabeling of a 5-min period as sleep for one of the flies. Thus, we conclude that the multibeam system is sensitive and accurate enough to detect micromovements that involve several body parts such as feeding and grooming, and that it is a valid tool to quantify sleep.

As seen in Fig 1M and 1N, 20 and 50 Hz optogenetic activations significantly increase sleep in *dFB-Split>CsChrimson* females as assessed using the DAM5H multibeam system, ruling out the possibility that we misregistered micromovements as sleep. However, to further support our findings, we performed video analysis. First, we coupled video and multibeam analysis of control and *dFB-Split>CsChrimson* females during 10 min of baseline recording and the subsequent 10 min of optogenetic activation at 20 Hz (Fig 1O). We found that upon optogenetic activation of dFB$^{23E10\cap84C10}$ neurons, walking is significantly reduced, and rest (total immobility with no micromovements) is increased (Fig 1P and Movies 3–16 available on https://osf.io/64zh9/?view_only=d058f62c75254556a22f11bd28979287). Importantly, we observed no change in micromovements upon optogenetic activation (Fig 1P and Movies 3–16 available on https://osf.io/64zh9/?view_only=d058f62c75254556a22f11bd28979287). Both multibeam and video analysis produced similar results, reinforcing the validity of the multibeam system to properly register sleep, movements, and micromovements (Fig 1P). Second, we performed video analysis of multiple behaviors before and during constant optogenetic activation of dFB$^{23E10\cap84C10}$ neurons and we found that times spent feeding, grooming, and walking are significantly reduced in *dFB-Split>CsChrimson* flies during activation, while time spent sleeping is significantly increased (Fig 1Q). Thus, based on our multibeam and video analysis,

we conclude that either a 20 Hz or a constant optogenetic activation of dFB$^{23E10∩84C10}$ neurons increases sleep, rather than micromovements.

Finally, we investigated whether activation of dFB$^{23E10∩84C10}$ neurons modulate arousal threshold by applying mechanical stimulations of increasing strength to *dFB-Split>CsChrimson* flies. As seen in Fig 1R, when dFB$^{23E10∩84C10}$ neurons are activated (retinal-fed, LED ON constant), a significantly lower percentage of flies are awakened by the 20% and 40% stimulus strength, compared with non-activated flies (retinal-fed, LED OFF). Importantly, at the higher stimuli strength (60%, 80%, and 100%), there are no differences between activated and non-activated *dFB-Split>CsChrimson* flies, indicating that these flies can respond to a stimulus of sufficient strength (i.e., these flies are not paralyzed). These data suggest that constant optogenetic activation of dFB$^{23E10∩84C10}$ neurons increases arousal threshold, an indicator of sleep depth.

Thus, considering our single beam, multibeam, and video analysis data, we conclude that optogenetic activation of dFB$^{23E10∩84C10}$ neurons at 10 Hz or above can increase sleep, sleep consolidation, and sleep depth.

## The sleep induced by dFB$^{23E10∩84C10}$ neuronal activation consolidates STM to LTM

To further support that activation of dFB$^{23E10∩84C10}$ neurons increases sleep and not micromovements, we sought to assess whether the state induced by activation of dFB$^{23E10∩84C10}$ neurons can support one of sleep's proposed functions. The relationship between sleep and memory is well documented [13,48,49]. In particular, sleep plays a role in the consolidation of STM into LTM [50]. Employing genetic and pharmacological activation, previous studies have demonstrated that inducing sleep after a courtship memory training protocol sufficient to create STM, but not LTM, could convert that STM into LTM in *Drosophila* [14,18,19]. Sleep depriving flies while activating sleep-promoting neurons abrogated the formation of LTM, demonstrating that sleep was needed for this effect [14]. More recent studies have shown that post-learning neuronal reactivation of dopaminergic neurons that are involved in memory acquisition is needed for LTM consolidation [18]. Importantly, this reactivation necessitates sleep and the activity of vFB neurons during a narrow time window after learning [18,51]. Based on their data, the authors proposed that vFB and dFB neurons promote sleep in response to different types of experiences and that these neurons underlie different functions of sleep [18]. We wondered whether the sleep induced by activation of dFB$^{23E10∩84C10}$ neurons is capable of converting STM to LTM. To test this, we used a 1 h courtship training protocol that does not create LTM and activated dFB$^{23E10∩84C10}$ neurons optogenetically for 23 h following the end of training. LTM was tested 24 h after the onset of training (Fig 2A). As seen in Fig 2B, sleep is significantly increased when activating dFB$^{23E10∩84C10}$ neurons post-training. As expected, control flies and *dFB-Split>CsChrimson* flies trained for 1 h, without increasing sleep post-training, show no LTM (Fig 2C). However, activating dFB$^{23E10∩84C10}$ neurons for 23 h after a 1 h courtship training session led to LTM, as indicated by a significantly higher suppression index (SI) (Fig 2C). Individual values of courtship indices for untrained and trained males are presented in S17A Fig. These data demonstrate that activation of dFB$^{23E10∩84C10}$ neurons following training can convert STM to LTM. An alternative hypothesis is that activation of dFB$^{23E10∩84C10}$ neurons promotes LTM consolidation independently of sleep. To rule out this possibility, we sleep-deprived *dFB-Split>CsChrimson* flies following the 1 h training session while they were optogenetically activated. This manipulation resulted in sleep loss (Fig 2B) and no LTM (Fig 2C). Thus, we conclude that the sleep that is induced by activation of dFB$^{23E10∩84C10}$ neurons can convert STM to LTM, supporting that the behavior observed is indeed sleep.

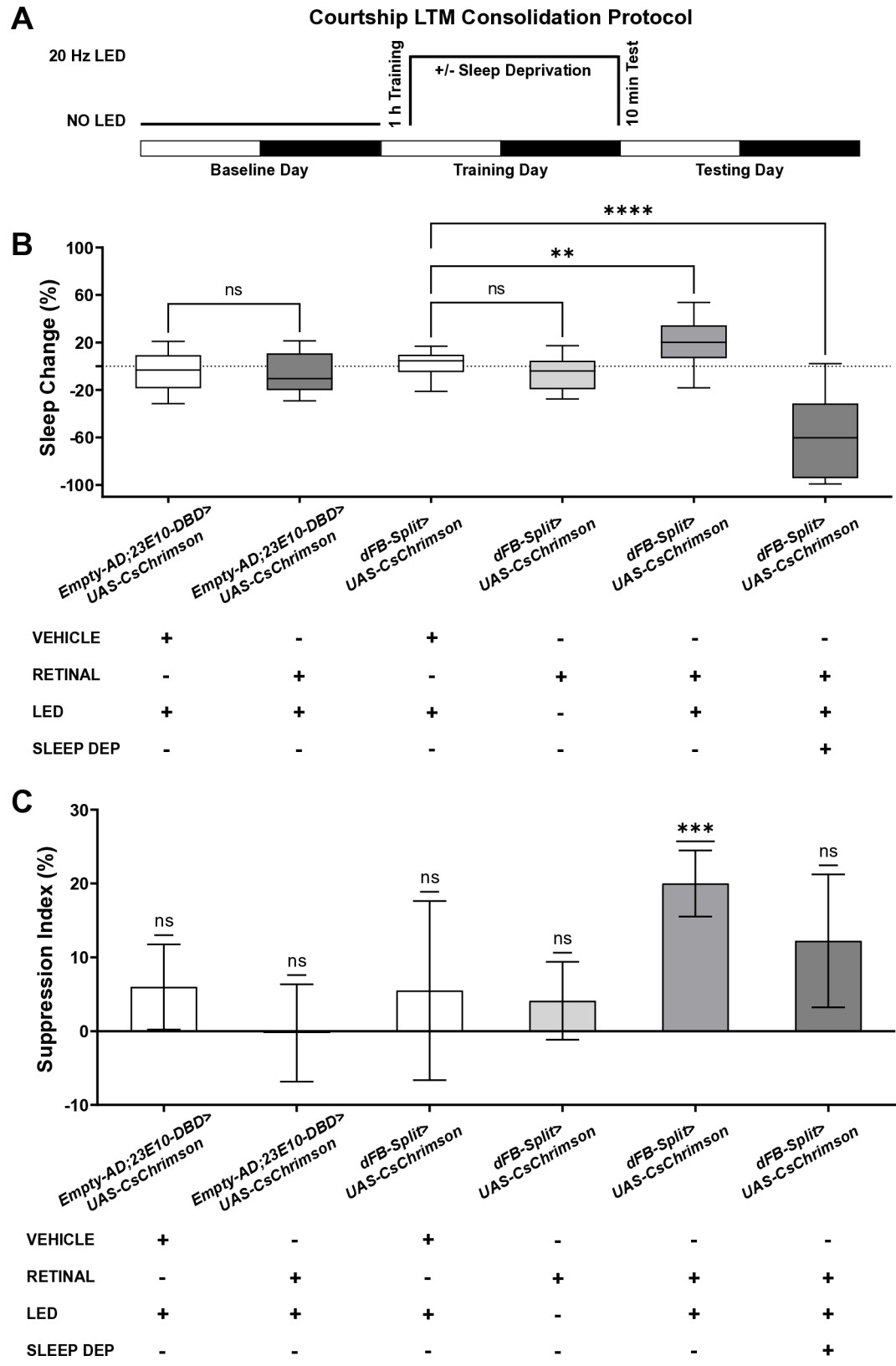

**Fig 2. The sleep induced by dFB[23E10∩84C10] neurons activation can consolidate LTM. (A)** Schematic of protocol for LTM consolidation. A 1 h training period that only generates STM is followed by optogenetic activation of dFB[23E10∩84C10] neurons for 23 h post-training (with or without sleep deprivation). Flies were tested 24 h after the onset of training. **(B)** Post-training

sleep change for different genotypes and condition. The ~23 h time period post-training was matched to the equivalent time period on the baseline day for comparison. Unpaired parametric *t* test for Empty-Split groups. Kruskal–Wallis test of multiple comparisons for dFB-Split groups using the vehicle condition as the control. ****$P < 0.0001$, **$P < 0.01$, ns = not significant. $n$ = 46-54 flies per genotype and condition. **(C)** Courtship LTM shown as the SI (SI = 100 * (1 – (CI Trained/ CI Untrained))) of trained fly groups tested 24 h after the onset of training. Wilcoxon signed-rank test using $H_0$, SI = 0. Sample size (untrained:trained) from left to right on the graph, $n$ = 54 (27:27), 51 (27:24), 49 (27:22), 51 (26:25), 46 (21:25), and 53 (27:26), respectively. Courtship indices <10% were excluded from both groups. ***$P < 0.001$, ns = not significant. The raw data underlying parts B and C can be found in S1 Data. dFB, dorsal fan-shaped body; LTM, long-term memory; SI, suppression index; STM, short-term memory.

## Silencing dFB$^{23E10 \cap 84C10}$ neurons increases sleep and reduces sleep homeostasis

After demonstrating that activation of dFB$^{23E10 \cap 84C10}$ neurons increases sleep duration and sleep depth, and that this sleep consolidates LTM, we performed experiments to silence the activity of these neurons chronically by expressing the hyperpolarizing inward rectifying potassium channel Kir2.1 [52]. First, we assessed whether the constitutive expression of Kir2.1 in dFB$^{23E10 \cap 84C10}$ neurons leads to anatomical defects. As seen in S18A and S18B Fig, we observed no gross morphological defects in dFB$^{23E10 \cap 84C10}$ neuron numbers or processes when expressing Kir2.1, compared with controls.

Surprisingly, total sleep is significantly increased when expressing Kir2.1 in dFB$^{23E10 \cap 84C10}$ neurons in female flies (Fig 3A). This effect on sleep is accompanied by an increase of daytime sleep bout duration (Fig 3C) while consolidation at night is unchanged (Fig 3D). These enhancements on sleep are not due to locomotor deficits (Fig 3B). We obtained identical behavioral results when expressing Kir2.1 in dFB$^{23E10 \cap 84C10}$ neurons of male flies (S18C–S18F Fig). These results are somewhat surprising as both chronic silencing and optogenetic activation of dFB$^{23E10 \cap 84C10}$ neurons result in sleep increases. We thus decided to investigate further. First, we repeated these behavioral experiments using the multibeam system. As seen in Fig 3E, chronic hyperpolarization of dFB$^{23E10 \cap 84C10}$ neurons increases total sleep as assessed with DAM5H monitors. Thus, chronic hyperpolarization of dFB$^{23E10 \cap 84C10}$ neurons increases sleep and sleep consolidation without disrupting the gross morphological properties of these cells. To further investigate the effects of silencing dFB$^{23E10 \cap 84C10}$ neurons, we employed an acute silencing approach by expressing Shi$^{ts1}$ [53]. Acute silencing (24 h, Fig 3F) of dFB$^{23E10 \cap 84C10}$ neurons has no effect on sleep in females (Fig 3G) or males (Fig 3H). Altogether, our chronic silencing data further demonstrate that dFB$^{23E10 \cap 84C10}$ neurons regulate sleep. However, it is puzzling that both activation and chronic silencing lead to sleep increases, and that acute silencing has no effect at all.

Multiple studies have suggested a role for dFB neurons in regulating sleep homeostasis. In particular, sleep deprivation increases the excitability of dFB neurons [16] and augmented sleep pressure switches dFB neurons from an electrically silent to an electrically active state [20]. Furthermore, reducing the excitability of dFB neurons by reducing levels of the Rho-GTPase-activating protein encoded by the crossveinless-c (cv-c) gene leads to a defect in sleep homeostasis [16]. We previously demonstrated that hyperpolarizing all 23E10-GAL4 neurons blocks sleep homeostasis; however, this effect was not due to VNC-SP neurons [27]. Thus, we concluded that there must be 23E10-GAL4 expressing neurons that are not VNC-SP cells that are involved in sleep homeostasis. We hypothesized that dFB$^{23E10 \cap 84C10}$ neurons could be the cells responsible for sleep homeostasis. To test this possibility, we expressed Kir2.1 in dFB$^{23E10 \cap 84C10}$ neurons, subjected flies to 12 h of mechanical sleep deprivation (SD) during the night and monitored recovery sleep for the subsequent 48 h. We observed that control flies show a gradual recovery of lost sleep over the 48 h period (Fig 3I). Hyperpolarizing dFB$^{23E10 \cap 84C10}$ neurons led to an interesting pattern as *dFB-Split>Kir2.1* flies show some

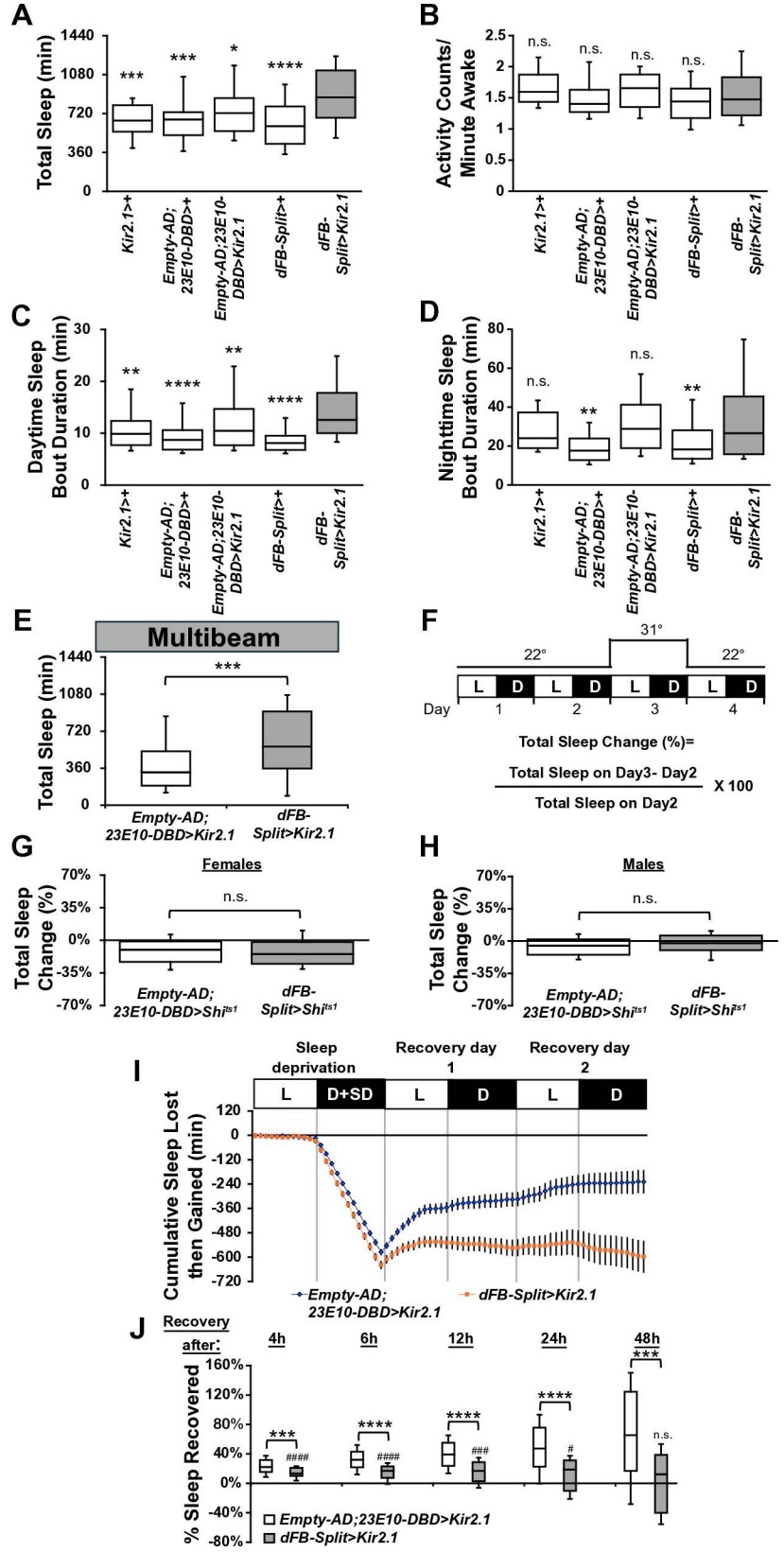

**Fig 3. Chronic hyperpolarization of dFB$^{23E10∩84C10}$ neurons promotes sleep and impairs sleep homeostasis. (A)** Box plots of total sleep (in minutes) for control and *dFB-Split>Kir2.1* female flies. A one-way ANOVA followed by Tukey's multiple comparisons revealed that *dFB-Split>Kir2.1* female flies sleep significantly more than controls. *$P <$

0.05, ***$P < 0.001$, ****$P < 0.0001$. $n = 40$-$85$ flies per genotype. **(B)** Box plots of locomotor activity counts per minute awake for flies presented in A. A Kruskal–Wallis ANOVA followed by Dunn's multiple comparisons revealed no differences between controls and *dFB-Split>Kir2.1* female flies. n.s. = not significant. $n = 40$-$85$ flies per genotype. **(C)** Box plots of daytime sleep bout duration (in minutes) for flies presented in A. A Kruskal–Wallis ANOVA followed by Dunn's multiple comparisons revealed that daytime sleep bout duration is increased in *dFB-Split>Kir2.1* female flies. ****$P < 0.0001$, **$P < 0.01$. $n = 40$-$85$ flies per genotype. **(D)** Box plots of nighttime sleep bout duration (in minutes) for flies presented in A. A Kruskal–Wallis ANOVA followed by Dunn's multiple comparisons revealed no differences between controls and *dFB-Split>Kir2.1* female flies. n.s. = not significant, **$P < 0.01$. $n = 40$-$85$ flies per genotype. **(E)** Box plots of total sleep (in minutes) for control and *dFB-Split>Kir2.1* female flies measured with the DAM5H multi-beam system. A two-tailed Mann–Whitney U test revealed that *dFB-Split>Kir2.1* female flies sleep significantly more than controls. ***$P < 0.001$. $n = 49$-$61$ flies per genotype. **(F)** Diagram of the experimental assay for acute silencing. Sleep was measured at 22 °C for 2 days to establish baseline sleep profile. Flies were then shifted to 31 °C for 24 h at the start of day 3 to silence the activity of the targeted cells by activating the $Shi^{ts1}$ actuator, and then returned to 22 °C on day 4. White bars (L) represent the 12 h of light and black bars (D) represent the 12 h of dark that are oscillating daily. **(G)** Box plots of total sleep change in % for control (*Empty-AD; 23E10-DBD>UAS-Shi*ts1), and *dFB-Split>UAS-Shi*ts1 female flies upon thermogenetic silencing. A two-tailed unpaired *t* test revealed no differences between controls and *dFB-Split>UAS-Shi*ts1 female flies. n.s. = not significant. $n = 50$ flies per genotype. **(H)** Box plots of total sleep change in % for control (*Empty-AD; 23E10-DBD>UAS-Shi*ts1), and *dFB-Split>UAS-Shi*ts1 male flies upon thermogenetic silencing. A two-tailed unpaired *t* test revealed no differences between controls and *dFB-Split>UAS-Shi*ts1 male flies. n.s. = not significant. $n = 60$-$62$ flies per genotype. **(I)** Cumulative sleep lost then gained for *Empty-AD; 23E10-DBD>UAS-Kir2.1* and *dFB-Split>UAS-Kir2.1* female flies during 12 h of mechanical sleep deprivation (D+SD) and 48 h of sleep recovery. **(J)** Box plots of total sleep recovered in % during 48 h of sleep recovery following 12 h of sleep deprivation at night for *Empty-AD; 23E10-DBD>UAS-Kir2.1* and *dFB-Split>UAS-Kir2.1* female flies. Two-tailed Mann–Whitney U tests revealed that sleep rebound is significantly decreased at all time points between controls and *dFB-Split>UAS-Kir2.1* female flies. However, Wilcoxon signed-rank tests (indicated by # on graph) using $H_{0:}$ % sleep recovered = 0 revealed that *dFB-Split>UAS-Kir2.1* female flies have a sleep rebound significantly greater than 0 at time points 4 h, 6 h, 12 h, and 24 h. ***$P < 0.001$, ****$P < 0.0001$, ####$P < 0.0001$, ###$P < 0.001$, #$P < 0.5$, n.s. = not significant. $n = 25$-$54$ flies per genotype. The raw data underlying parts A, B, C, D, E, G, H, I, and J can be found in S1 Data. AD, activation domain; DBD, DNA-binding domain; dFB, dorsal fan-shaped body.

recovery during the first 4–6 h following SD, but then did not show additional gains during the remaining recovery period (Fig 3I). To further demonstrate our finding, we quantified sleep recovery after 4 h, 6 h, 12 h, 24 h, and 48 h. As seen in Fig 3J, sleep recovery is significantly reduced between controls and *dFB-Split>Kir2.1* flies at all time points. However, in *dFB-Split>Kir2.1* flies, sleep recovery at 4 h, 6 h, 12 h, and 24 h is significantly different from 0, indicating that these flies show a weaker, but not nonexistent, homeostatic rebound (Fig 3J). These data suggest that hyperpolarizing dFB[23E10∩84C10] neurons do not fully block sleep homeostasis, especially in the immediate period following sleep deprivation. Interestingly, another work showed that hyperpolarizing all 23E10-GAL4 neurons does not impair early (6 h) sleep homeostasis [28]. In conclusion, our data suggests that there may be multiple groups of neurons involved in sleep homeostasis in the fly brain, with dFB[23E10∩84C10] neurons being an important, but not the sole, player.

## dFB[23E10∩84C10] neurons express acetylcholine and glutamate

Previous studies have proposed that dFB neurons are GABAergic [54,55]. However, these data were obtained using a driver that is not dFB-specific [27]. In addition, some dFB neurons transcribe both the vesicular glutamate transporter (VGlut) and the vesicular acetylcholine transporter (VAChT) [56]. Furthermore, we showed in our previous work [27] that some 23E10-GAL4 dFB neurons are cholinergic. Taken all together, these data indicate that the neurochemical identity of dFB neurons is uncertain. We thus investigated whether dFB neurons are GABAergic, glutamatergic, or cholinergic by expressing GFP in dFB[23E10∩84C10] neurons and staining them with antibodies to GABA, choline acetyltransferase (ChAT, the enzyme necessary to produce acetylcholine), and VGlut. As seen in Fig 4A, we observed no GABA staining in dFB[23E10∩84C10] neurons, but instead found that a minority of these cells express

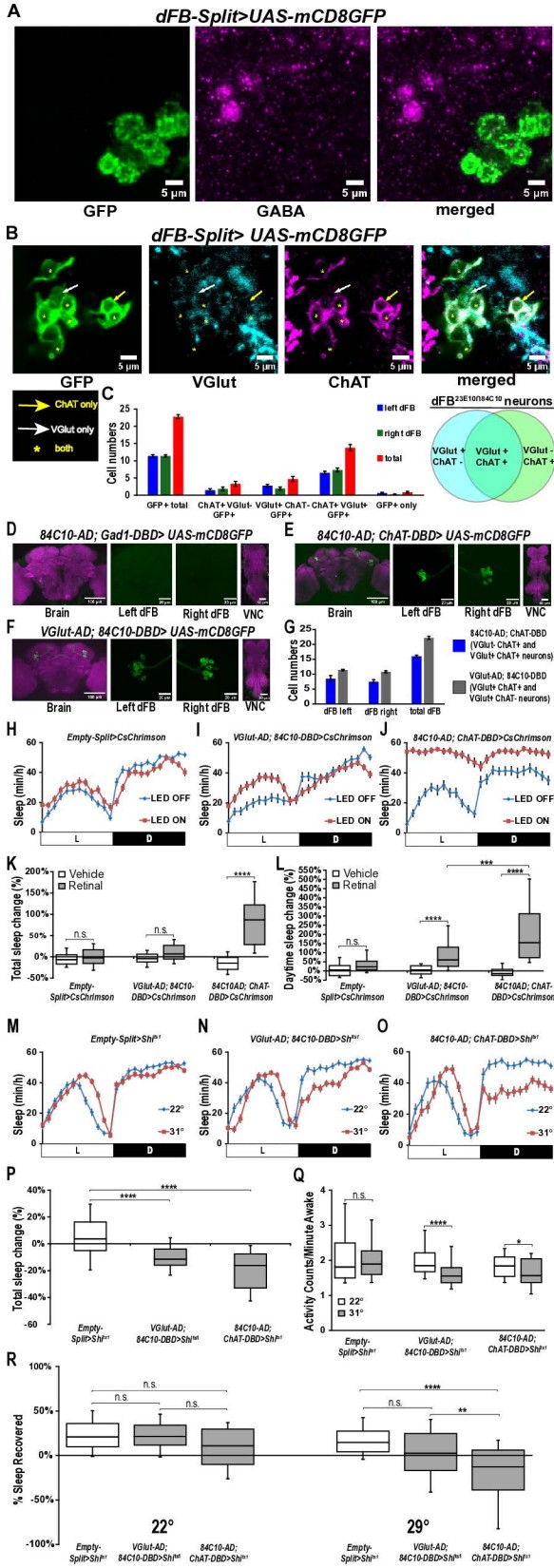

**Fig 4. Neurochemical identity of dFB²³ᴱ¹⁰ⁿ⁸⁴ᶜ¹⁰ neurons.** (A) Representative confocal stack of a female *dFB-Split>UAS-mCD8GFP* brain stained with GFP and GABA antibodies and focusing on dFB²³ᴱ¹⁰ⁿ⁸⁴ᶜ¹⁰ cell bodies.

Green, anti-GFP; magenta, anti-GABA. **(B)** Representative confocal stack of a female *dFB-Split>UAS-mCD8GFP* brain stained with GFP, VGlut and ChAT antibodies and focusing on dFB$^{23E10 \cap 84C10}$ cell bodies. White arrow indicates a VGlut only positive dFB$^{23E10 \cap 84C10}$ neurons. Yellow arrows show ChAT only positive dFB$^{23E10 \cap 84C10}$ neurons. Yellow asterisks indicate dFB$^{23E10 \cap 84C10}$ neurons positive for VGlut and ChAT. Green, anti-GFP; gray, anti-VGlut, magenta, anti-ChAT. **(C)** Quantification of VGlut only positive, ChAT only positive and VGlut and ChAT positive dFB$^{23E10 \cap 84C10}$ neurons. We observed $22.76 \pm 0.58$ ($n = 13$) GFP positive dFB$^{23E10 \cap 84C10}$ neurons per brain, $3.31 \pm 0.71$ cells were ChAT only positive, $4.69 \pm 0.73$ cells were VGlut only positive, $13.77 \pm 0.93$ cells were ChAT and VGlut positive, and $0.85 \pm 0.25$ cells were negative for both ChAT and VGlut. **(D)** Representative confocal stack of a female *84C10-AD; Gad1-DBD>UAS-mCD8GFP*. Green, anti-GFP; magenta, anti-nc82 (neuropile marker). **(E)** Representative confocal stack of a female *84C10-AD; ChAT-DBD>UAS-mCD8GFP*. Green, anti-GFP; magenta, anti-nc82 (neuropile marker). **(F)** Representative confocal stack of a female *VGlut-AD; 84C10-DBD>UAS-mCD8GFP*. Green, anti-GFP; magenta, anti-nc82 (neuropile marker). **(G)** Quantification of the numbers of dFB neurons contained within the 2 Split-GAL4 lines presented in E and F. We observed $16.00 \pm 0.41$ ($n = 4$) dFB cells for *84C10-AD; ChAT-DBD* and $22.20 \pm 0.49$ ($n = 5$) dFB neurons for *VGlut-AD; 84C10-DBD*. No other expression was seen in the brain and VNC for these lines. **(H)** Sleep profile in minutes of sleep per hour for day 2 (LED OFF, blue line) and day 3 (LED ON, red line) for retinal-fed Empty-Split control females expressing CsChrimson subjected to a 50 Hz optogenetic activation protocol (cycles of 5 ms LED ON, 15 ms LED OFF). **(I)** Sleep profile in minutes of sleep per hour for day 2 (LED OFF, blue line) and day 3 (LED ON, red line) for retinal-fed *VGlut-AD; 84C10-DBD>CsChrimson* female flies subjected to a 50 Hz optogenetic activation protocol (cycles of 5 ms LED ON, 15 ms LED OFF). **(J)** Sleep profile in minutes of sleep per hour for day 2 (LED OFF, blue line) and day 3 (LED ON, red line) for retinal-fed *84C10-AD; ChAT-DBD>CsChrimson* female flies subjected to a 50 Hz optogenetic activation protocol (cycles of 5 ms LED ON, 15 ms LED OFF). **(K)** Box plots of total sleep change in % ((total sleep on activation day-total sleep on baseline day/total sleep on baseline day) × 100) for control (*Empty-Split>CsChrimson*), *VGlut-AD; 84C10-DBD>CsChrimson* and *84C10-AD; ChAT-DBD>CsChrimson* female flies under a 50 Hz optogenetic activation protocol. Two-way ANOVA followed by Sidak's multiple comparisons revealed that activating 84C10-AD; ChAT-DBD neurons significantly increases sleep. n.s. = not significant, ****$P < 0.0001$. $n = 32$-$47$ flies per genotype and condition. **(L)** Box plots of daytime sleep change in % ((daytime sleep on activation day-daytime sleep on baseline day/daytime sleep on baseline day) × 100) for control (*Empty-Split>CsChrimson*), *VGlut-AD; 84C10-DBD>CsChrimson* and *84C10-AD; ChAT-DBD>CsChrimson* female flies under a 50 Hz optogenetic activation protocol. Two-way ANOVA followed by Tukey's multiple comparisons revealed that activating 84C10-AD; ChAT-DBD and VGlut-AD; 84C10-DBD neurons significantly increases daytime sleep and that activating 84C10-AD; ChAT-DBD neurons increase sleep more than activation of VGlut-AD; 84C10-DBD cells. n.s. = not significant, ***$P < 0.001$, ****$P < 0.0001$. $n = 32$-$47$ flies per genotype and condition. **(M)** Sleep profile in minutes of sleep per hour for *Empty-Split>Shi*ts1 females maintained at 22° (baseline) and 31° (activation). **(N)** Sleep profile in minutes of sleep per hour for *VGlut-AD; 84C10-DBD>Shi*ts1 females maintained at 22° (baseline) and 31° (activation). **(O)** Sleep profile in minutes of sleep per hour for *84C10-AD; ChAT-DBD>Shi*ts1 females maintained at 22° (baseline) and 31° (activation). **(P)** Box plots of total sleep change in % ((total sleep on activation day-total sleep on baseline day/total sleep on baseline day) × 100) for control (*Empty-Split>Shi*ts1), *VGlut-AD; 84C10-DBD>Shi*ts1 and *84C10-AD; ChAT-DBD>Shi*ts1 female flies. A Kruskal–Wallis ANOVA followed by Dunn's multiple comparisons revealed that acutely silencing VGlut-AD; 84C10-DBD and 84C10-AD; ChAT-DBD neurons significantly decreases sleep. ****$P < 0.0001$. $n = 26$-$60$ flies per genotype. **(Q)** Box plots of locomotor activity counts per minute awake for flies presented in P. Two-way repeated measures ANOVA followed by Sidak's multiple comparisons revealed that acutely silencing VGlut-AD; 84C10-DBD and 84C10-AD; ChAT-DBD neurons does not lead to hyperactivity. n.s. = not significant, *$P < 0.05$, ****$P < 0.0001$. $n = 26$-$60$ flies per genotype. **(R)** Box plots of total sleep recovered in % during 24 h of recovery following 12 h of sleep deprivation at night for *Empty-Split>Shi*ts1, *VGlut-AD; 84C10-DBD>Shi*ts1 and *84C10-AD; ChAT-DBD>Shi*ts1 female flies. Following sleep deprivation, flies were either maintained at 22° (control) or at 31° (to silence neurons). Kruskal–Wallis test followed by Dunn's multiple comparisons revealed that acutely silencing 84C10-AD; ChAT-DBD neurons blocks sleep homeostasis. No differences were seen between lines at 22°. n.s. = not significant, **$P < 0.01$, ****$P < 0.0001$. $n = 28$-$57$ flies per genotype and condition. The raw data underlying parts C, G, K, L, P, Q, and R can be found in S1 Data. AD, activation domain; ChAT, choline acetyltransferase; DBD, DNA-binding domain; dFB, dorsal fan-shaped body; VGlut, vesicular glutamate transporter; VNC, ventral nerve cord.

ChAT only or VGlut only, while more than 50% of them express both neurotransmitters (Fig 4B and 4C). Note that while this manuscript was in revision, a new preprint demonstrated that dFB neurons express VGlut, VAChT, and ChAT [57], in agreement with our findings and previously published data [56]. Our immunohistochemical analysis thus uncovered that dFB$^{23E10 \cap 84C10}$ neurons can be divided in 3 subgroups: neurons that express ChAT and VGlut (ChAT$^+$, VGlut$^+$ cells), neurons that express only VGlut (ChAT$^-$, VGlut$^+$ cells), and neurons that express only ChAT (ChAT$^+$, VGlut$^-$ cells) (Fig 4C, right).

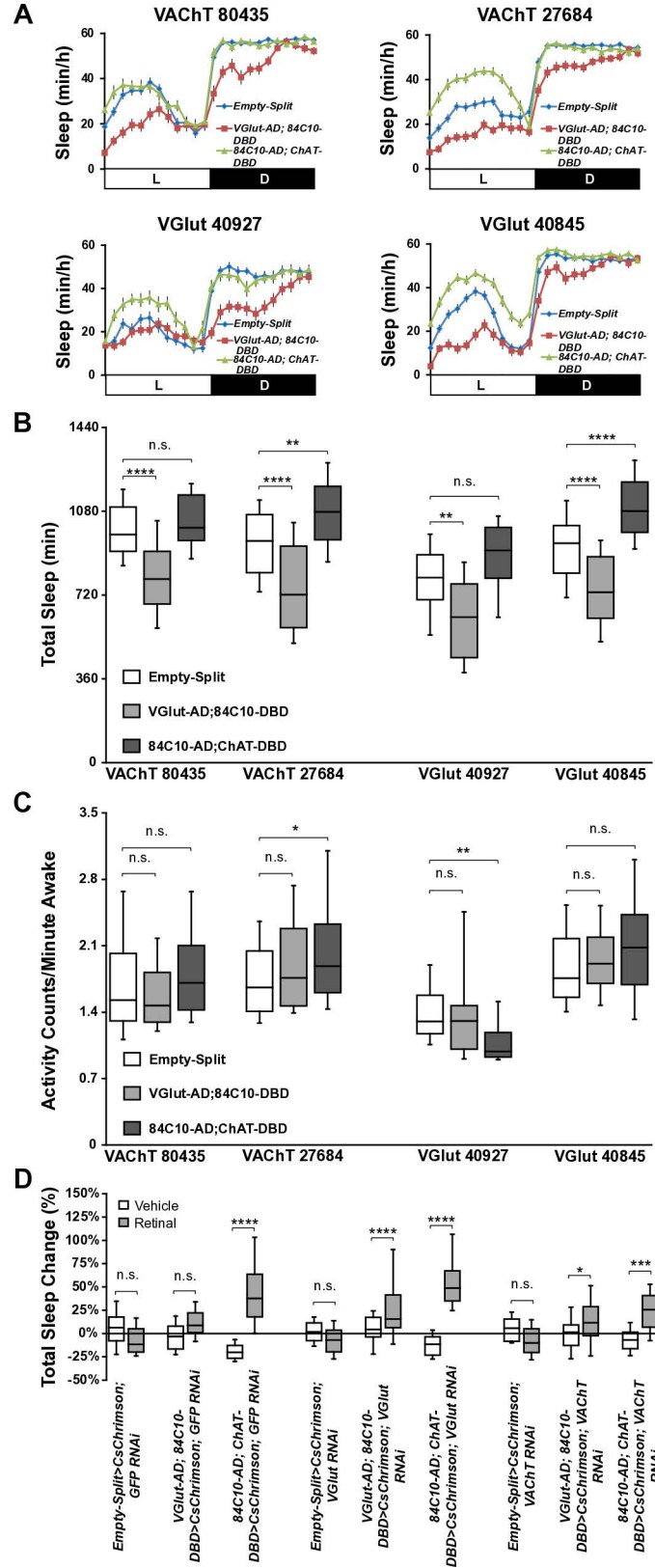

**Fig 5. Cholinergic and glutamatergic transmission in dFB neurons. (A)** Sleep profile in minutes of sleep per hour for Empty-Split control, VGlut-AD; 84C10-DBD and 84C10-AD; ChAT-DBD females expressing RNAi against VAChT (80435 and 27684) and VGlut (40927 and 40845). **(B)** Box plots of total sleep (in minutes) for flies presented

in A. One-way ANOVA followed by Tukey's multiple comparisons for each individual RNAi line. n.s. = not signifi-cant, **$P < 0.01$, ****$P < 0.0001$. $n = 20$-68 flies per genotype. **(C)** Box plots of locomotor activity counts per minute awake for flies presented in A. Kruskal–Wallis ANOVA followed by Dunn's multiple comparisons for each individ-ual RNAi line. n.s. = not significant, *$P < 0.05$, **$P < 0.01$. $n = 20$-68 flies per genotype. **(D)** Box plots of total sleep change in % for female control (Empty-Split), VGlut-AD; 84C10-DBD, and 84C10-AD; ChAT-DBD flies expressing CsChrimson and RNAi against GFP (control), VAChT (line 27684) or VGlut (line 40845) under 50 Hz activation. Two-way ANOVA followed by Sidak's multiple comparisons for each individual RNAi line. n.s. = not significant, *$P < 0.05$, ***$P < 0.001$, ****$P < 0.0001$. $n = 15$-43 flies per genotype and condition. The raw data underlying parts B, C, and D can be found in S1 Data. AD, activation domain; DBD, DNA-binding domain; dFB, dorsal fan-shaped body; VAChT, vesicular acetylcholine transporter.

To confirm that 23E10-GAL4 dFB neurons are not GABAergic, we expressed GFP using 23E10-GAL4 and stained brains with an antibody to GABA. As seen in S19A Fig, we observed no GABA staining in 23E10-GAL4 dFB neurons. In addition, both a Gad1-AD; 23E10-DBD and 84C10-AD; Gad1-DBD Split-GAL4 lines show no expression in dFB neurons (Figs 4D and S19B). Thus, we conclude that dFB$^{23E10 \cap 84C10}$ neurons are not GABAergic, contrary to previous reports. To strengthen our findings about the glutamatergic and cholinergic nature of dFB$^{23E10 \cap 84C10}$ neurons, we employed a Split-GAL4 strategy using the 84C10 component of dFB-Split paired with neurotransmitter-specific Split elements. An 84C10-AD; ChAT-DBD line expresses in about 16 dFB neurons (ChAT$^+$, VGlut$^+$ and ChAT$^+$, VGlut$^-$ cells; named dFB$^{ChAT \cap 84C10}$) per brain and no other cells in the brain and VNC (Fig 4E and 4G and S3 and S4 Movies). A VGlut-AD; 84C10-DBD line expresses in 22 dFB cells (ChAT$^+$, VGlut$^+$ and ChAT$^-$, VGlut$^+$ cells; named dFB$^{VGlut \cap 84C10}$) per brain and shows no other expression in the brain and VNC (Fig 4F and 4G and S5 and S6 Movies). Note that while this manuscript was in prepara-tion, a preprint reported that dFB neurons are glutamatergic and that optogenetic activation of these cells weakly, but significantly, increases sleep [58]. However, the Split-GAL4 line used in this study, VGlut-AD; 23E10-DBD may not be dFB-specific, making the interpretation of the behavioral data difficult. We have created 2 different VGlut-AD; 23E10-DBD lines using the only 2 VGlut-AD lines known to us. As seen in S19C and S19D Fig and S7 and S8 Movies, both lines express in dFB neurons, but they also express in 8 neurons that are not dFB cells. Note that both independent lines show similar expression in these 8 non-dFB neurons. Thus, our data suggest that VGlut-AD; 23E10-DBD is not a dFB-specific tool and cannot be used to manipulate only dFB neurons. Taken together, our immunostaining data and Split-GAL4 approach indicate that dFB$^{23E10 \cap 84C10}$ neurons are not homogenous when it comes to neuro-chemical identity and can be divided in 3 categories. Considering that acetylcholine is the main excitatory neurotransmitter in the fly system and that glutamate is inhibitory in dFB neurons [58], in addition to the high level of recurrent connectivity in the dFB [58,59], it is reasonable to assume that these different categories of dFB$^{23E10 \cap 84C10}$ neurons may differentially affect sleep.

To investigate this possibility, we employed optogenetic activation of dFB$^{VGlut \cap 84C10}$ and dFB$^{ChAT \cap 84C10}$ neurons. While these tools are more homogeneous than the dFB-Split line (which contains the 3 populations of dFB neurons), they are not perfect, as both express in 2 of the 3 dFB neurons populations. When we employed our regular optogenetic activation protocol, we saw no effects, perhaps unsurprisingly based on the data obtained when activat-ing all dFB$^{23E10 \cap 84C10}$ neurons (S20A Fig). We then increased the frequency of the optogenetic stimulation to 20 Hz and found that activating dFB$^{ChAT \cap 84C10}$ neurons significantly increased sleep while no effects were observed with dFB$^{VGlut \cap 84C10}$ cells (S20B Fig). A 50 Hz optoge-netic activation of dFB$^{ChAT \cap 84C10}$ neurons strongly increases total sleep while it does not with dFB$^{VGlut \cap 84C10}$ neurons (Fig 4H–K). However, when only examining daytime sleep, we found

that a 50 Hz activation of dFB$^{VGlut\Omega 84C10}$ neurons significantly increases sleep, although significantly less than activation of the cholinergic neurons (Fig 4L). Sleep is not increased during the night when activating dFB$^{VGlut\Omega 84C10}$ neurons (S20C Fig). Both daytime and nighttime sleep are increased when activating dFB$^{ChAT\Omega 84C10}$ neurons (Figs 4L and S20C). These increases in sleep are not caused by a general deficit in locomotor activity (S20F Fig). These data suggest that within the heterogeneous dFB population, cholinergic neurons play a major role in the sleep-promoting capacity of this region. We obtained similar effects using the multi-beam system, ruling out that we have falsely classified periods of micromovements as sleep (S21A–S21D Fig). Further inspection of sleep architecture revealed that while activation of dFB$^{VGlut\Omega 84C10}$ neurons increases daytime sleep, it does so by increasing the number of sleep episodes that flies initiate during the day rather than sleep consolidation (S20D and S20E Fig). During the night, sleep bout numbers are increased, but sleep consolidation is reduced (S20G and S20H Fig). Thus, it appears that activation of dFB$^{VGlut\Omega 84C10}$ neurons increases sleep initiation but does not promote sleep consolidation. This is in marked contrast to the activation of dFB$^{ChAT\Omega 84C10}$ neurons, which increases sleep consolidation during day and night (S20E and S20H Fig). None of these effects are observed in vehicle-fed controls (S20I–S20M Fig). These data suggest that dFB$^{VGlut\Omega 84C10}$ and dFB$^{ChAT\Omega 84C10}$ neurons are likely modulating different aspects of sleep and position dFB$^{ChAT\Omega 84C10}$ neurons as the strongest modulators of sleep and sleep consolidation within the dFB.

We then acutely silenced the activity of dFB$^{VGlut\Omega 84C10}$ and dFB$^{ChAT\Omega 84C10}$ neurons by expressing Shi$^{ts1}$. As seen in Fig 4M–P, silencing either set of dFB neurons significantly reduces sleep, particularly at night. These effects are not due to hyperactivity (Fig 4Q). Altogether, these data confirm that dFB$^{VGlut\Omega 84C10}$ and dFB$^{ChAT\Omega 84C10}$ neurons regulate sleep. Finally, we investigated homeostatic sleep in flies in which dFB$^{VGlut\Omega 84C10}$ and dFB$^{ChAT\Omega 84C10}$ neurons were silenced during the recovery period. As seen in Fig 4R, there were no differences in sleep homeostasis between controls, *VGlut-AD; 84C10-DBD>Shi$^{ts1}$*, and *84C10-AD; ChAT-DBD>Shi$^{ts1}$* flies when maintained at 22° post-sleep deprivation. However, raising the temperature to 29° to activate Shi$^{ts1}$ led to an abrogation of sleep rebound in flies with silenced dFB$^{ChAT\Omega 84C10}$ neurons while not affecting sleep rebound in flies with silenced dFB$^{VGlut\Omega 84C10}$ neurons (Fig 4R).

## Cholinergic and glutamatergic signaling in the dFB

To investigate whether cholinergic and glutamatergic transmission plays a role in the sleep modulatory capacity of the dFB, we expressed RNAi constructs against VGlut and VAChT in dFB$^{VGlut\Omega 84C10}$ and dFB$^{ChAT\Omega 84C10}$ neurons. As seen in Fig 5A and 5B, expressing 2 RNAi lines against VAChT and 2 RNAi lines against VGlut in dFB$^{VGlut\Omega 84C10}$ neurons significantly reduces sleep in female flies. These effects are not caused by hyperactivity (Fig 5C). For dFB$^{ChAT\Omega 84C10}$ neurons, we found that expressing one VAChT RNAi line (27684) or one VGlut RNAi line (40845) significantly increases sleep (Fig 5A and 5B), and that these effects are not due to a locomotor defect (Fig 5C). The second VGlut line (40927) increases sleep during the daytime but it also leads to hypoactivity (Fig 5A–C), making conclusions about its use difficult.

These data not only suggest that both cholinergic and glutamatergic signaling act in dFB neurons to regulate sleep, but also highlight differences between dFB$^{VGlut\Omega 84C10}$ and dFB$^{ChAT\Omega 84C10}$ neurons. Interestingly, a recent preprint reported that reducing VGlut levels in all 23E10-GAL4 expressing neurons reduces sleep [58]. However, whether these phenotypes can be attributed to dFB neurons is uncertain.

To investigate whether cholinergic and glutamatergic transmission is necessary for the sleep-promoting capacity of dFB$^{VGlut\Omega 84C10}$ and dFB$^{ChAT\Omega 84C10}$ neurons when optogenetically activated, we expressed RNAi constructs against GFP (control), VAChT, and VGlut, while also expressing CsChrimson. As seen in Fig 5D, we found that reducing VAChT or VGlut levels in

dFB$^{ChAT\Omega84C10}$ neurons does not block the sleep increase observed when using a 50 Hz optogenetic activation in female flies. There are multiple possible explanations for these findings. First, a subset of dFB$^{ChAT\Omega84C10}$ neurons also express glutamate (ChAT$^+$, VGlut$^+$ cells), so it is possible that these neurons use either glutamatergic or cholinergic transmission to downstream targets. This would explain why reducing levels of only one of these 2 single vesicular transporters cannot block the sleep increase obtained with optogenetic activation. Secondly, it is possible that there is enough vesicular transporter remaining in our RNAi experiments, enabling signaling to downstream targets. Alternatively, dFB$^{ChAT\Omega84C10}$ neurons may use another neuromodulator to promote sleep. Given that the central complex, which contains the dFB, is one of the most peptidergic areas of the fly brain [60,61], we think that it is extremely likely that dFB neurons are peptidergic. In fact, a recent preprint has identified many peptides expressed in the dFB [57]. For dFB$^{VGlut\Omega84C10}$ neurons, we observed no increase in total sleep when expressing GFP RNAi and CsChrimson, in agreement with our findings expressing only CsChrimson (Fig 4K). However, when expressing both VAChT or VGlut RNAi in addition to CsChrimson, we found that a 50 Hz optogenetic activation of dFB$^{VGlut\Omega84C10}$ neurons significantly increases sleep (Fig 5D). These data suggest that VGlut and VAChT may reduce the sleep-promoting capacity of dFB$^{VGlut\Omega84C10}$ neurons and that these cells may also use another neuromodulator to increase sleep when sufficiently activated.

## Discussion

The strength of the fly model lies in its strong genetic techniques, allowing researchers to selectively manipulate discrete populations of neurons and monitor how specific behaviors are affected. To do so, an impressive collection of binary expression systems has been developed. In particular, the GAL4/UAS system [3] has been the keystone of *Drosophila* neurobiological studies. However, GAL4 drivers are often not specific enough and express in cells outside the region of interest. This can make the task of unequivocally assigning a given behavior to a specific neuron or group of neurons particularly difficult. This lack of specificity can be addressed by employing the intersectional Split-GAL4 technology to refine GAL4 expression [6].

Like mammalian sleep, sleep in *Drosophila* is regulated by multiple areas in the brain [12,13]. An expanding number of studies have suggested a role for dFB neurons in sleep regulation [14–17,20,22,24,55,62]. The most widely used tool to manipulate dFB neurons is the 23E10-GAL4 driver [20,21], and thermogenetic or optogenetic activation of 23E10-GAL4 neurons results in increased sleep [18,24–27]. However, we previously demonstrated that 23E10-GAL4 expresses in 2 cholinergic sleep-promoting neurons located in the VNC (VNC-SP neurons) [27], putting the role of 23E10-GAL4 dFB neurons in question. The involvement of VNC-located neurons in 23E10-GAL4 sleep-promotion has been demonstrated further by a subsequent paper [28]. Simply put, we think that 23E10-GAL4 should no longer be employed as a dFB-specific driver line. Two recent studies, using 23E10-GAL4, suggested that the dFB plays no role at all in sleep [28,29], a conclusion we did not make based on the finding that there are 23E10-GAL4 expressing neurons involved in sleep homeostasis that are different from VNC-SP neurons [27]. However, whether these sleep homeostasis-regulating 23E10-GAL4 neurons are dFB neurons was uncertain. In this study, we sought to assess the role of 23E10-GAL4 dFB neurons in sleep regulation.

To do so, we adopted a targeted Split-GAL4 strategy combining 20 individual AD lines (selected for the strong dFB expression pattern of their GAL4 counterparts) with a 23E10-DBD line. We screened these 20 novel Split-GAL4 FBS lines behaviorally and anatomically. These experiments identified 4 FBS lines that strongly promote sleep when activated in females and 7 in males (S2 Table). In addition, we found 9 lines that moderately modulate at least 1 sleep parameter (S2 Table). Interestingly, only 4 FBS lines do not change any sleep

parameter using both thermogenetic and optogenetic activation protocols (S2 Table). An anatomical assessment of each line revealed that all sleep-promoting FBS lines express in dFB neurons (S1 Table), as expected based on our targeted approach. However, they also express in the metathoracic ganglion of the VNC, especially in the previously described VNC-SP neurons [27]. The high frequency of observation of these cells in our FBS lines, and their very typical "bowtie" processes in the brain, suggest that VNC-SP neurons are commonly expressed in many GAL4 lines. In fact, a MultiColor FlpOut (MCFO) study reported that neurons with similar projections (referred to as "sparse T" in this work) are observed in more than 60% of all Janelia GAL4 lines [63]. The fact that these "bowtie" neurons are frequently present in many GAL4 lines probably explains why we see them in the vast majority of our FBS lines (17 out of 20, or 85% of them). This observation also highlights the need to use specific tools when trying to manipulate discrete neurons or groups of neurons. Since VNC-SP neurons are present in many GAL4 lines, it is likely that they are part of the expression pattern of GAL4 lines that have been used to manipulate diverse groups of sleep-modulating neurons. Future studies using GAL4 lines in sleep studies will therefore need to take account of the potential presence of VNC-SP neurons in their expression pattern. Perhaps more importantly, existing data may need to be reinterpreted in light of our findings. Our study also found that the strength of expression within VNC-SP neurons dictates how potent a given FBS line is in promoting sleep. However, the ubiquitous presence of VNC-SP cells in sleep-promoting FBS lines prevented us from unequivocally assessing the role of dFB neurons in sleep regulation using these lines.

Fortunately, we identified a dFB-specific Split-GAL4 line (dFB-Split), which expresses in most 23E10-GAL4 dFB neurons (dFB$^{23E10 \cap 84C10}$ neurons). Using either a 1 Hz or our regular optogenetic protocol, activation of dFB$^{23E10 \cap 84C10}$ neurons does not modulate sleep. However, when we increased the intensity of optogenetic activation (10 Hz or above), we found that activating dFB$^{23E10 \cap 84C10}$ neurons increases sleep in both males and females. Importantly, we demonstrated this sleep-induction effect using the single beam DAM2 system, the multi-beam DAM5H monitor, and by video analysis, ruling out that we have mislabeled periods of micromovements, grooming, or feeding as sleep. In addition, the state induced by activation of dFB$^{23E10 \cap 84C10}$ neurons consolidates LTM (see below), suggesting that this state is indeed sleep. Finally, we showed that activation of dFB$^{23E10 \cap 84C10}$ neurons increases arousal threshold, indicating that flies sleep deeper when dFB$^{23E10 \cap 84C10}$ neurons are activated. Crucially, though the sleep obtained by dFB$^{23E10 \cap 84C10}$ neurons activation is deeper, it is reversible, which is an important hallmark of sleep.

Chronic hyperpolarization of dFB$^{23E10 \cap 84C10}$ neurons interferes with sleep homeostasis. We found that expressing Kir2.1 in dFB$^{23E10 \cap 84C10}$ neurons leads to a reduction in the amount of sleep that flies recover following sleep deprivation, when compared with controls. However, *dFB-Split>Kir2.1* flies show a modest, but significant sleep rebound, especially during the first hours in the recovery period. Interestingly, a recent study found that hyperpolarizing all 23E10-GAL4 neurons does not impair early (6 h) sleep homeostasis [28]. These data demonstrate that dFB$^{23E10 \cap 84C10}$ neurons are involved in sleep homeostasis but indicate that there are other sleep homeostasis-regulating neurons in the fly nervous system, a suggestion also made by others [64].

Surprisingly, chronic silencing of dFB$^{23E10 \cap 84C10}$ neurons increases sleep, confirming that dFB neurons modulate sleep. This effect is not caused by gross morphological defects created by the expression of Kir2.1 throughout the development and life of the flies. How do we explain this result? dFB neurons have been mostly implicated in sleep homeostasis. To be efficient at inducing sleep only in response to sleep pressure, the activity of the underlying neuronal system must be tightly regulated. This is the case for dFB neurons, which are subject

to strong dopaminergic regulation [15,17,20,45] and are highly interconnected [58,59]. Additionally, dFB[23E10∩84C10] neurons are neurochemically heterogeneous (see below), containing 3 different groups of neurons based on their expression of glutamate and acetylcholine. Acetylcholine is the main excitatory neurotransmitter in the fly nervous system, while glutamate acts as an inhibitory neurotransmitter in dFB neurons [58]. Thus, the dFB is a highly interconnected network containing excitatory and inhibitory connections that is under strong dopaminergic inhibition under baseline conditions. This organization probably ensures stability within the dFB network, to regulate its activity in response to increasing sleep pressure effectively. We hypothesize that chronic hyperpolarization of all dFB[23E10∩84C10] neurons disrupts the balance of inhibitory and excitatory connections within the dFB. This may lead to complex effects on the activity of individual cells in this heterogeneous population, generating unexpected and complicated sleep phenotypes. Using neurotransmitter- and dFB-specific Split-GAL4 lines to acutely activate or silence more homogeneous populations of dFB neurons led to sleep increases and sleep decreases, respectively.

Why do dFB[23E10∩84C10] neurons require such a strong activation protocol to increase sleep? Considering that dFB neurons are subject to dopaminergic regulation [15,17,20,45], we hypothesize that under baseline conditions, dFB neurons are under severe dopaminergic inhibition and that our regular optogenetic activation protocol is not sufficient to bring these cells above firing threshold. Increasing the intensity of the activation protocol above 10 Hz reliably leads to sleep increases in what appears to be a direct relationship. Since chronic silencing and optogenetic activation of dFB[23E10∩84C10] neurons both increase sleep, a potential issue is that our 24 h-long optogenetic activation protocols lead to neuronal fatigue, which would phenocopy hyperpolarization. However, we feel that this is unlikely for multiple reasons. First, following 12 h of sleep deprivation, individual dFB neurons increase their firing frequencies, with some neurons capable of firing up to 50 Hz [16]. These data indicate that our optogenetic protocols, using 10, 20, and 50 Hz frequencies to activate dFB neurons, are within the physiological range of normal firing properties for these cells. Second, when using neurotransmitter-specific dFB Split-GAL4 lines (see below), we show that activation of these dFB neurons promote sleep while acute silencing leads to wake. If sustained activation of these dFB neurons led to neuronal fatigue, we would expect to obtain the same behavioral effect as silencing, which is not the case. Rather, we explain the sleep increase seen when chronically hyperpolarizing all dFB[23E10∩84C10] neurons by a disruption of the balance between inhibitory and excitatory connections within the dFB.

A recent preprint showed that optogenetic activation of dFB neurons, as labeled with a VGlut-AD; 23E10-DBD line, increased sleep when using an activation protocol consisting of 10 bursts of LED ON compressed in one half of a 500:500 ms duty cycle. Interestingly, a 10 Hz activation, delivering the same number of light pulses spread evenly across time failed to increase sleep [58]. The authors thus concluded that the temporal structure of the activation protocol used to activate dFB neurons is important to their sleep-promoting capacities [58]. Using our dFB-specific tool, we show that using optogenetic activations of 10 Hz or more lead to significant increases in sleep. Thus, we propose a simpler model, in which the intensity of the activation protocol dictates whether or not dFB neurons promote sleep when activated. While we expect that the expression pattern of our dFB-Split driver and the VGlut-AD; 23E10-DBD line contains mostly similar neurons (since they both rely on 23E10-DBD), we show in this study that they are not fully identical. Our analysis, using 2 independent VGlut-AD; 23E10-DBD lines, shows that while both lines express in many dFB neurons, they also express in 8 additional non-dFB neurons. Thus, VGlut-AD; 23E10-DBD is not a suitable tool to assess the role of the dFB in sleep regulation, as it is not a dFB-specific driver line. We hypothesize that these differences in expression pattern probably explain the difference between our results and others [58].

Previous work showed that dFB neurons are heterogeneous [16] and proposed that these neurons are GABAergic [54,55]. Our work, using both Split-GAL4 and immunocytochemistry approaches, reveals that dFB neurons are not GABAergic. We believe that the mischaracterization of dFB neurons as GABAergic is a consequence of using a non-dFB-specific tool, 23E10-GAL4. This finding further reinforces the need to use specific tools when trying to decipher the neural circuitries underlying complex behaviors.

In this study, we found that most dFB$^{23E10\cap84C10}$ neurons express both glutamate and acetylcholine, while a minority of them only express one of these 2 neurotransmitters. These data are in agreement with a previous study [56] and a recent preprint [57]. Thus, dFB$^{23E10\cap84C10}$ neurons can be divided into 3 subgroups: ChAT$^{+}$, VGlut$^{+}$ cells; ChAT$^{-}$, VGlut$^{+}$ cells; and ChAT$^{+}$, VGlut$^{-}$ cells. Using neurotransmitter- and dFB-specific tools, our behavioral data support a major role in sleep promotion and sleep homeostasis for the cholinergic dFB$^{ChAT\cap84C10}$ neurons, which consist of 2 of the 3 dFB subgroups. Activating glutamatergic dFB$^{VGlut\cap84C10}$ neurons has a much milder effect on sleep, only increasing sleep during the day. When examining sleep architecture, we found that dFB$^{ChAT\cap84C10}$ neurons increase sleep consolidation when activated, while dFB$^{VGlut\cap84C10}$ cells do not. We note that activation of VGlut-AD; 23E10-DBD expressing neurons also led to a modest sleep increase during the day [58]; however, since this driver is not dFB-specific, the cellular origin of this sleep increase is uncertain. Acute silencing of either dFB$^{VGlut\cap84C10}$ or dFB$^{ChAT\cap84C10}$ neurons reduces sleep, further demonstrating that dFB neurons regulate sleep.

Our RNAi analysis revealed that dFB neurons rely on cholinergic and glutamatergic signaling to modulate sleep, as reducing levels of VGlut or VAChT in dFB$^{VGlut\cap84C10}$ and dFB$^{ChAT\cap84C10}$ neurons modulates sleep in opposite directions. These RNAi data further support that dFB neurons regulate sleep but may be difficult to interpret, as these manipulations are targeting 2 of the 3 dFB subgroups. For example, expressing VGlut RNAi in dFB$^{VGlut\cap84C10}$ neurons affects the ChAT$^{+}$, VGlut$^{+}$ cells and ChAT$^{-}$, VGlut$^{+}$ cells, but expressing VAChT RNAi should only affect the ChAT$^{+}$, VGlut$^{+}$ neurons. When considering the high level of interconnectivity within the dFB [58,59], predicting the sleep effect of RNAi expression in multiple subgroups of dFB neurons may be difficult. This means that further dissection of the dFB circuitry is required to fully elucidate how this neuronal population functions. Nevertheless, we hypothesize that reducing the levels of VGlut or VAChT in different groups of dFB neurons disrupts the balance of excitatory and inhibitory connections within the dFB, mitigating its stability, and leading to sleep phenotypes under baseline conditions.

Our RNAi analysis also suggests that in addition to cholinergic and glutamatergic signaling, dFB neurons may use another neuromodulator to increase sleep when activated. We suspect that dFB neurons use neuropeptidergic transmission to increase sleep. This is based on the fact that the central complex, which contains the dFB is highly peptidergic [57,60,61]. In fact, a previous study proposed that dFB neurons express the Allatostatin-A (AstA) neuropeptide [55], the fly homolog of galanin, an important sleep-regulating peptide in the mammalian brain [65]. However, a second study claims that there is no AstA expression in the dFB [28]. Whether AstA or another neuropeptide regulates the sleep-promoting capacity of dFB$^{VGlut\cap84C10}$ and dFB$^{ChAT\cap84C10}$ neurons requires further studies. Taken together, these data indicate that additional genetic dissection and studies of individual dFB neurons will be needed to fully understand the contribution of neurochemically distinct dFB neurons to sleep regulation.

Finally, a previous study demonstrated that vFB neurons regulate learning-dependent sleep and that these cells play an essential role in LTM consolidation [18]. The same work proposed that vFB and dFB neurons promote different types of sleep that may underlie different functions [18]. While vFB and dFB neurons may promote sleep in response to different drives, a learning-dependent sleep drive for vFB neurons and a homeostatic sleep drive for dFB cells,

we found that the sleep obtained by activation of dFB$^{23E10 \cap 84C10}$ neurons can consolidate LTM. Thus, we conclude that the benefit of sleep on memory consolidation is not circuit specific.

In conclusion, our work demonstrates that the dFB is neurochemically heterogeneous and that these neurons modulate sleep and sleep homeostasis. In addition, dFB-generated sleep can consolidate LTM. Further work is needed to dissect the heterogeneous dFB population and will identify the connectivity and contribution of individual dFB neurons in sleep regulation.

## Materials and methods

### *Drosophila* stocks and rearing

Flies were cultured at 25 °C with 50% humidity under a 12 h light:12 h dark cycle. Flies were kept on a standard yeast and molasses diet (per 1 L: 50 g yeast, 15 g sucrose, 28 g corn syrup, 33.3 ml molasses, 9 g agar). Fly stocks used in this study are listed in S3 Table.

### Sleep behavioral experiments

Sleep was assessed as previously described [11,19,27], and 4- to 10-day-old virgin females or males were used as described in figure legends. Briefly, flies were placed into individual 65 mm tubes and all activity was continuously measured through the Trikinetics Drosophila Activity Monitoring System (DAM2, www.Trikinetics.com, Waltham, Massachusetts, United States of America). Locomotor activity was measured in 1-min bins and sleep was defined as periods of quiescence lasting at least 5 min. For multibeam monitoring, we used the DAM5H monitors (www.Trikinetics.com, Waltham, Massachusetts, USA).

### Multibeam system validation: Paired MB and video analysis

To validate the accuracy and sensitivity of the Trikinetics DAM5H multibeam system, Canton-S flies were placed into individual 65 mm tubes and monitored by the DAM5H system while simultaneously being video recorded. The multibeam system was set to record moves (fly movement from one beam to another), counts (fly movement within a beam), and position in the tube. Flies were monitored and recorded for 10 min. The multibeam data was measured in 1-min bins and each minute was scored based on highest activity level that occurred in that minute: moves, counts, and rest. A 1-min bin that had moves was scored as moves. A 1-min bin that had counts, but no moves, was scored as micromovements. A 1-min bin that had neither moves nor counts was scored as rest. The videos were manually analyzed for fly behaviors that all fell into the different movement categories of moves (walking), micromovements (feeding, grooming, proboscis extension, posture change, single leg movement), and rest. These behaviors were scored in the same 1-min bins to compare directly with multibeam detection. Sensitivity and accuracy of the multibeam were calculated by the following formulas:

Sensitivity = minutes with a behavior identified by the multibeam system/ all visually labeled minutes of the same behavior

Accuracy = correctly identified minutes with a behavior by the multibeam system to/ all identified minutes with the same behavior by the multibeam system

### Multibeam and video analysis of optogenetic activation of control and dFB-Split flies

For Fig 1P, flies expressing CsChrimson were loaded into individual 65 mm tubes with food supplemented with 400 μm all-trans retinal and monitored by the DAM5H system while

simultaneously being video recorded. The multibeam system was set to record moves (fly movement from one beam to another), counts (fly movement within a beam), and position in the tube. Flies were monitored and recorded for 10 min of baseline activity with LEDs OFF followed immediately by 10 min of activation with LEDs ON at 20 Hz. Multibeam data and video analysis was performed in the same manner as for multibeam validation with *Canton-S* flies.

### Quantitative video analysis

For Fig 1Q, video recording was performed as previously described [27]. Flies fed food supplemented with 400 μm all-trans retinal (Sigma, #R2500) or vehicle were loaded into individual 65 mm tubes and placed on custom-built platforms. One webcam for 2 platforms was used to record the flies for 30 min starting at ZT1 (1 h after lights turn on in the morning). Fly behavior was manually scored and categorized as walking, grooming, feeding, resting (inactivity < 5 min), or sleeping (inactivity > 5 min). Videos were recorded on consecutive days for baseline (LED OFF) and activation (LED ON). Three separate experiments were performed and the pooled data is shown.

### Optogenetic activation

For optogenetic activation, flies expressing CsChrimson were loaded into individual 65 mm tubes with food supplemented with 400 μm all-trans retinal (Sigma, #R2500) or vehicle control and kept under 12 h low-light/12 h darkness conditions for 48 h. Sleep was then monitored for 24 h of baseline measurements. To activate CsChrimson, flies were put under 627 nm LEDs (LuxeonStar LXM2-PD01–0040) and set to different activation protocols. The regular intensity protocol is set to a pulse cycle of (5 ms on, 95 ms off) × 20 with a 4 s delay between pulse cycles. The light intensity of this protocol is 0.078 mW/mm$^2$. For the 1 Hz protocol, a cycle of 5 ms on, 995 ms off flashing continuously was used (light intensity = 0.05 mW/mm$^2$). For the 10 Hz protocol a cycle of 5 ms on, 95 ms off flashing continuously was used (light intensity = 0.078 mW/mm$^2$). For the 20 Hz protocol a cycle of 5 ms on, 45 ms off flashing continuously was used (light intensity = 0.118 mW/mm$^2$). For 50 Hz, a cycle of 5 ms on, 15 ms off flashing continuously was used (light intensity = 0.252 mW/mm$^2$). The light intensity of the LED constantly ON protocol is 0.565 mW/mm$^2$. A photodiode sensor (S120C, Thorlabs) paired with a digital console (PM100D, Thorlabs) was used to measure light intensity.

### Sleep deprivation assay

Sleep deprivation was performed as previously described [19,27,66,67]. Briefly, flies were placed into individual 65 mm tubes and the sleep-nullifying apparatus (SNAP) was used to sleep deprive flies for 12 h during the dark phase. For Fig 3J, sleep homeostasis was calculated for each individual fly as the ratio of the minutes of sleep gained above baseline after 4 h, 6 h, 12 h, 24 h, and 48 h of recovery sleep divided by the total min of sleep lost during 12 h of sleep deprivation. For Fig 4R, flies were maintained at 22° during baseline and sleep deprivation. Following 12 h of sleep deprivation, a group of flies was maintained at 22° during recovery while another one was transferred to 29° to activate Shi[ts1]. Sleep homeostasis was calculated for each individual fly as the ratio of the minutes of sleep gained above baseline during 24 h of recovery sleep divided by the total min of sleep lost during 12 h of sleep deprivation.

### Arousal threshold

Arousal threshold was measured on flies that were optogenetically activated. Control flies and flies expressing CsChrimson were loaded into individual 65 mm glass tubes containing food

supplemented with 400 μm all-trans retinal (Sigma, #R2500) or vehicle control. After 3 days, the tubes were placed onto custom 3D printed trays, 16 tubes per tray. Video of 2 trays was recorded using a Raspberry Pi NoIR camera. Arousal stimulus was provided by three coreless vibration motors (7 × 25 mm) capable of 8,000–16,000 rmp vibration. Vibration pattern and intensity were controlled by a DRV2605 haptic controller (Adafruit) paired with a Raspberry Pi 3 Model B+. A custom program was written in python to control the video and motor vibration. The programmed sequence starts with turning video recording on. Ten minutes after recording starts the stimulus pattern begins. The DRV2605 vibration effect used is transition click (effects 62–58) starting at 20% intensity and increasing in 20% increments to 100% intensity. Vibration occurred for 200 ms followed by 800 ms no vibration repeating 4 times followed by a 15 s break before the next increasing stimulus starts. Video recording continued for 10 min after final vibration stimulus. This program repeated once per hour from ZT3 to ZT9. Videos were manually scored. Flies inactive during the 5 min directly before the stimulus began were scored as sleeping. Sleeping flies were then assessed for movement during the increasing intensity stimulus. Arousal was reported based on the percentage of flies moving at each vibration intensity. Arousal threshold was scored on consecutive days for baseline (LED OFF) and activation (LED ON). The code for the arousal threshold can be found at https://github.com/mcellinj/Dissel-lab-arousal-apparatus.

## Immunocytochemistry

Flies were cold anesthetized, brains and VNC were dissected in ice cold Schneider's *Drosophila* Medium (Gibco, 21720024), and blocked in 5% normal goat serum. Following blocking, flies were incubated overnight at 4 °C in primary antibody, washed in PBST, and incubated overnight at 4 °C in secondary antibody. Primary antibodies used were chicken anti-GFP (1:1,000; Aves Labs, Inc, #GFP-1020), mouse anti-bruchpilot (1:50; Developmental Studies Hybridoma Bank (DSHB), nc82-s), mouse anti-ChAT (1:100; DSHB, Chat4b1-c), rabbit anti-vGlut (1:500; DiAntonio lab [68], Washington University), and rabbit anti-GABA (1:1,000; Sigma-Aldrich, #A2052). Secondary antibodies used were goat anti-chicken AlexaFluor 488 (1:800; Invitrogen, #A32931), goat anti-mouse AlexaFluor 555 (1:400; Invitrogen #A32727), goat anti-rabbit AlexaFluor 555 (1:400; Invitrogen, #A32732), goat anti-mouse AlexaFluor 633 (1:400; Invitrogen, $A21052), and goat anti-rabbit AlexaFluor 633 (1:400; Invitrogen, #A21071). Brains and VNCs were mounted on polylysine-treated slides in Vectashield H-1000 mounting medium. Imaging was performed on a Zeiss 510 meta confocal microscope using a Plan-Apochromat 20× or Plan-Neofluar 40× objective. Z-series images were acquired with a 1 μm slice size using the same settings (laser power, gain, offset) to allow for comparison across genotypes. Images were processed and analyzed using ImageJ.

For non-CNS adult tissues (wings, legs, ovaries, and guts). Flies were collected, fixed, and stained similarly as brains and VNCs but following incubation with secondary antibodies and washing 3 times with PBST, tissue was incubated with 10 μm 4′,6-Diamidino-2-Phenylindole, Dihydrochloride (DAPI) (Invitrogen D1306) for 15 min at room temperature. Tissues were then washed in PBST and mounted in Vectashield.

## GFP-DD

For destabilized GFP (GFP-DD) experiments, flies expressing GFP-DD were maintained on standard food. Flies were transferred to food containing 1 mM Trimethoprim (TMP) solution (T1225, Teknova) or DMSO as control. Following 24 h on TMP or DMSO containing food, flies were cold anesthetized and brains and VNCs were dissected. Brains and VNCs were fixed in 4% paraformaldehyde, washed in PBST, and mounted on slides. Brains and VNCs were

imaged on a Zeiss 510 meta confocal microscope using a Plan-Neofluar 40× objective. GFP intensity was measured in ImageJ.

## Courtship conditioning protocol

Naïve males were collected upon eclosion and isolated into glass sleep tubes; 48+ hours prior to training, males were transferred into new glass sleep tubes containing either vehicle or all-trans retinal dissolved in food and loaded into monitors. The 24 h period prior to training served as the baseline sleep measurement. Naïve males were transferred into 13 mm × 100 mm glass culture tubes with food at the bottom, with respective vehicle or all-trans retinal additions, and paired for 1 h with a mated trainer female (trained group) or alone (untrained group). Males were then transferred back into sleep tubes on the respective diet and sleep was monitored for the 23 h post-training period. Flies subjected to LED activation were placed under 627 nm LED light at 20 Hz (5 ms ON and 45 ms OFF continuous cycles) for the post-training period. The sleep deprived group experienced LED activation and sleep deprivation over the 23 h post-training period using the SNAP [19,27,66,67], and 24 h after the onset of training, male flies were placed into courtship chambers with a mated tester female and video recorded for the 10 min testing period. The courtship index (CI) was calculated using CI = (Time Spent on Courtship Behaviors/ Total Time of Test) * 100. The SI was then calculated using SI = 100 * (1 – (CI Trained/ CI Untrained)). Trainer and tester female mating status was visually confirmed after *don juan*-GFP pairing prior to training.

## Courtship video processing and scoring

Courtship videos were recorded for 10 min at 20 fps through OBS Studio software. Videos were analyzed using FlyTracker software [69,70] and outputs were saved as JAABA files. The visualization tool within FlyTracker was used to confirm the assignment of male and female flies and any switches in labeling were corrected using this feature. The videos and corresponding output files were scored using JAABA, a MATLAB-based machine learning program [71]. Positive and negative frame examples were used to train 3 different scoring classifiers: OriChase (includes orientation and chase), SingWing (singing and wing extension), and MountCop (includes mounting and copulation). The accuracy of these classifiers was verified by built-in ground truth data and comparison to hand-scored tests. JAABAPlot was used to output the CI for each male fly and the resulting SI was calculated from these values.

## Statistical analysis

Statistical analyses were performed with Prism10 software (GraphPad). Normal distribution was assessed with the D'Agostino–Pearson test. Normally distributed data were analyzed with parametric statistics: *t* test, one-way analysis of variance or two-way ANOVA followed by the planned pairwise multiple comparisons as described in the legends. For data that significantly differed from the normal distribution, non-parametric statistics were applied, including Mann–Whitney U test, Kruskal–Wallis test followed by Dunn's multiple test, and Wilcoxon signed-rank test. Some non-normally distributed data were subjected to log transformation or Box-Cox transformation before two-way ANOVA followed by planned pairwise multiple comparisons as described in the legends. When box plots are used, the bottom and top of each box represents the first and third quartile and the horizontal line dividing the box is the median. The whiskers represent the 10th and 90th percentiles. All statistically different groups are defined as *$P < 0.05$, **$P < 0.01$, ***$P < 0.001$, and ****$P < 0.0001$.

## Supporting information

**S1 Fig. Behavioral and anatomical screen of FBS lines. (A)** Diagram of the experimental assay. Sleep was measured at 22 °C (for thermogenetic activation) or with LED OFF (for the optogenetic activation protocol) for 2 days to establish baseline sleep profile. Flies were then shifted to 31 °C (thermogenetic) or LEDs were turned ON (LED ON, optogenetic) for 24 h at the start of day 3 to increase activity of the targeted cells by activating the TrpA1 or CsChrimson channel, and then returned to 22 °C or LED OFF on day 4. White bars (L) represent the 12 h of light and black bars (D) represent the 12 h of dark that are oscillating daily. **(B)** Box plots of total sleep change in % ((total sleep on day 3-total sleep on day 2/total sleep on day 2) × 100) for female control (Empty: Empty-AD; 23E10-DBD) and 20 FBS lines expressing UAS-TrpA1; UAS-mCD8GFP. The bottom and top of each box represents the first and third quartile, and the horizontal line dividing the box is the median. The whiskers represent the 10th and 90th percentiles. The gray rectangle spanning the horizontal axis indicates the interquartile range of the control. Kruskal–Wallis ANOVA followed by Dunn's multiple comparisons revealed that 4 FBS lines increase sleep significantly more than control flies when thermogenetically activated. $*P < 0.05$, $**P < 0.01$, $***P < 0.001$, $****P < 0.0001$, $n = 30–45$ flies per genotype. **(C)** Box plots of total sleep change in % ((total sleep on day 3-total sleep on day 2/total sleep on day 2) × 100) for vehicle-fed and retinal-fed control (Empty) and 20 FBS female flies expressing CsChrimson upon 627 nm LED stimulation. The bottom and top of each box represents the first and third quartile, and the horizontal line dividing the box is the median. The whiskers represent the 10th and 90th percentiles. Two-way ANOVA followed by Sidak's multiple comparisons revealed that 6 retinal-fed FBS lines increase sleep significantly when stimulated with 627 nm LEDs when compared with vehicle-fed flies. $**P < 0.01$, $***P < 0.001$, $****P < 0.0001$, $n = 20–40$ flies per genotype and condition. **(D–H)** Left, sleep profile in minutes of sleep per hour for day 2 (22 °C, blue line) and day 3 (31 °C, red line) for empty control (Empty-AD; 23E10-DBD, **D**), FBS42 (**E**), FBS45 (**F**), FBS53 (**G**) and FBS68 (**H**) female flies expressing TrpA1 and mCD8GFP. Right, Representative confocal stacks for each *FBS>UAS-TrpA1; UAS-mCD8GFP* line of a female brain (left panel), VNC (middle panel) as well as a magnified view of the location of VNC-SP "bowtie" processes. Green, anti-GFP; magenta, anti-nc82 (neuropile marker). The raw data underlying parts B and C can be found in S1 Data.
(TIF)

**S2 Fig. Additional sleep profiles and confocal images of FBS lines. (A–P)** Sleep profile (left) and representative confocal stacks (right) for brain and VNC of female *FBS>UAS-TrpA1; UAS-mCD8GFP* for 16 FBS lines not presented in S1 Fig.
(TIF)

**S3 Fig. Additional sleep parameters for FBS female flies thermogenetically activated. (A)** Box plots of locomotor activity counts per minute awake for flies presented in S1B Fig. The bottom and top of each box represents the first and third quartile, and the horizontal line dividing the box is the median. The whiskers represent the 10th and 90th percentiles. Two-way repeated measures ANOVA followed by Sidak's multiple comparisons test found that for 2 sleep-promoting FBS lines (FBS45 and FBS53) locomotor activity per awake time is increased while no differences are seen for the other 2 sleep-promoting lines between 22 and 31 °C. $**P < 0.01$, $****P < 0.0001$, $n = 30–45$ flies per genotype. **(B)** Box plots of daytime sleep bout duration in minutes for flies presented in S1B Fig. Two-way repeated measures ANOVA followed by Sidak's multiple comparisons test revealed that 7 FBS lines show a significant increase in daytime sleep bout duration between 22 and 31 °C. $*P < 0.05$, $**P < 0.01$, $***P < 0.001$, $****P < 0.0001$, $n = 30–45$ flies per genotype. **(C)** Box plots of nighttime sleep bout

duration in minutes for flies presented in S1B Fig. Two-way repeated measures ANOVA followed by Sidak's multiple comparisons test revealed that control and most FBS lines show a significant decrease in nighttime sleep bout duration between 22 and 31 °C. Only 4 sleep-promoting FBS lines show no difference between 22 and 31 °C. Dunnett's multiple comparisons reveal that for 3 of them, nighttime sleep bout duration at 31 °C is significantly increased compared with Empty control flies (# on figure). $**P < 0.01$, $***P < 0.001$, $****P < 0.0001$, n.s. = not significant. $n = 30–45$ flies per genotype. The raw data underlying parts A, B, and C can be found in S1 Data.
(TIF)

**S4 Fig. Thermogenetic activation in male flies.** **(A)** Box plots of total sleep change in % ((total sleep on day 3-total sleep on day 2/total sleep on day 2) × 100) for male control (Empty-AD; 23E10-DBD) and 20 FBS lines expressing UAS-TrpA1; UAS-mCD8GFP. The gray rectangle spanning the horizontal axis indicates the interquartile range of the control. Kruskal–Wallis ANOVA followed by Dunn's multiple comparisons revealed that 7 FBS lines increase sleep significantly more than control flies when transferred to 31 °C. $*P < 0.05$, $****P < 0.0001$, $n = 26–47$ flies per genotype. **(B)** Box plots of locomotor activity counts per minute awake for flies presented in A. Two-way repeated measures ANOVA followed by Sidak's multiple comparisons test found that for 3 sleep-promoting FBS lines (FBS45, FBS53, and FBS68) locomotor activity per awake time is increased while no differences are seen for the other 4 sleep-promoting lines between 22 and 31 °C. $*P < 0.05$, $**P < 0.01$, $***P < 0.001$, $****P < 0.0001$, $n = 26–47$ flies per genotype. **(C)** Box plots of daytime sleep bout duration for flies presented in A. Two-way repeated measures ANOVA followed by Sidak's multiple comparisons test found that for 6 out of the 7 sleep-promoting FBS lines, daytime sleep bout duration is significantly increased at 31 °C compared with 22 °C. $**P < 0.01$, $***P < 0.001$, $****P < 0.0001$, $n = 26–47$ flies per genotype. **(D)** Box plots of nighttime sleep bout duration for flies presented in A. Two-way repeated measures ANOVA followed by Sidak's multiple comparisons revealed that control and most FBS lines show a significant decrease in nighttime sleep bout duration between 22 °C and 31 °C, and 4 sleep-promoting FBS lines show no difference between 22 and 31 °C (FBS42, FBS53, FBS81, and FBS84) while FBS45 and FBS68 show an increase in nighttime sleep bout duration at 31 °C. $**P < 0.01$, $****P < 0.0001$, $n = 26–47$ flies per genotype. The raw data underlying parts A, B, C, and D can be found in S1 Data.
(TIF)

**S5 Fig. Additional sleep profiles for FBS lines in optogenetic screen. Sleep profiles in minutes of sleep per hour for vehicle-fed and retinal-fed FBS lines in optogenetic screen.**
(TIF)

**S6 Fig. Optogenetic activation in males** . **(A)** Box plots of total sleep change in % ((total sleep on day 3-total sleep on day 2/total sleep on day 2) × 100) for control (Empty) and 20 experimental vehicle-fed and retinal-fed male flies expressing CsChrimson upon 627 nm LED stimulation. Two-way ANOVA followed by Sidak's multiple comparisons revealed that 8 retinal-fed FBS lines increase sleep significantly when stimulated with 627 nm LEDs when compared with vehicle-fed flies. $*P < 0.05$, $**P < 0.01$, $****P < 0.0001$. $n = 20–44$ flies per genotype and condition. **(B)** Box plots of locomotor activity counts per minute awake for retinal-fed flies presented in A. Two-way repeated measures ANOVA followed by Sidak's multiple comparisons test found that for most sleep-promoting FBS lines, locomotor activity per awake time is not affected when the flies are stimulated with 627 nm LEDs while it is increased in *FBS72>UAS-CsChrimson* flies. $*P < 0.05$, $n = 24–44$ flies per genotype. **(C)** Box plots of daytime sleep bout duration (in minutes) for retinal-fed flies presented in A. Two-way repeated measures ANOVA followed by Sidak's multiple comparisons indicate that daytime sleep bout

duration is increased in 11 FBS lines expressing CsChrimson when stimulated with 627 nm LEDs. *$P < 0.05$, ***$P < 0.001$, ****$P < 0.0001$, $n = 24$–44 flies per genotype. **(D)** Box plots of nighttime sleep bout duration (in minutes) for retinal-fed flies presented in A. Two-way repeated measures ANOVA followed by Sidak's multiple comparisons indicate that nighttime sleep bout duration is increased in 6 FBS lines expressing CsChrimson when stimulated with 627 nm LEDs. **$P < 0.01$, ****$P < 0.0001$, $n = 24$–44 flies per genotype. The raw data underlying parts A, B, C, and D can be found in S1 Data.
(TIF)

**S7 Fig. Additional data for retinal-fed females in optogenetic activation. (A)** Box plots of locomotor activity counts per minute awake for retinal-fed flies presented in S1C Fig. The bottom and top of each box represents the first and third quartile, and the horizontal line dividing the box is the median. The whiskers represent the 10th and 90th percentiles. Two-way repeated measures ANOVA followed by Sidak's multiple comparisons test found that for most sleep-promoting FBS lines, locomotor activity per awake time is not affected when the flies are stimulated with 627 nm LEDs while it is increased in *FBS45>UAS-CsChrimson* flies. ****$P < 0.0001$, $n = 21$–40 flies per genotype. **(B)** Box plots of daytime sleep bout duration (in minutes) for retinal-fed flies presented in S1C Fig. The bottom and top of each box represents the first and third quartile, and the horizontal line dividing the box is the median. The whiskers represent the 10th and 90th percentiles. Two-way repeated measures ANOVA followed by Sidak's multiple comparisons indicate that daytime sleep bout duration is increased in 9 FBS lines expressing CsChrimson when stimulated with 627 nm LEDs. *$P < 0.05$, **$P < 0.01$, ***$P < 0.001$, ****$P < 0.0001$, $n = 21$–40 flies per genotype. **(C)** Box plots of nighttime sleep bout duration (in minutes) for retinal-fed flies presented in S1C Fig. The bottom and top of each box represents the first and third quartile, and the horizontal line dividing the box is the median. The whiskers represent the 10th and 90th percentiles. Two-way repeated measures ANOVA followed by Sidak's multiple comparisons indicate that nighttime sleep bout duration is increased in 6 FBS lines expressing CsChrimson when stimulated with 627 nm LEDs. *$P < 0.05$, ****$P < 0.0001$, $n = 21$–40 flies per genotype. The raw data underlying parts A, B, and C can be found in S1 Data.
(TIF)

**S8 Fig. Vehicle-fed sleep data for females in optogenetic experiments. (A)** Box plots of locomotor activity counts per minute awake for vehicle-fed flies presented in S1C Fig. Two-way repeated measures ANOVA followed by Sidak's multiple comparisons test found that all lines except FBS42 show no difference in locomotor activity per awake time when the flies are stimulated with 627 nm LEDs. ***$P < 0.001$. $n = 20$–39 flies per genotype. **(B)** Box plots of daytime sleep bout duration for vehicle-fed flies presented in S1C Fig. Two-way repeated measures ANOVA followed by Sidak's multiple comparisons test found that most vehicle-fed sleep-promoting lines show no difference in daytime sleep bout duration when the flies are stimulated with 627 nm LEDs. *$P < 0.05$, **$P < 0.01$, ***$P < 0.001$. $n = 20$–39 flies per genotype. **(C)** Box plots of nighttime sleep bout duration for vehicle-fed flies presented in S1C Fig. Two-way repeated measures ANOVA followed by Sidak's multiple comparisons test show no difference in nighttime sleep bout duration when vehicle-fed flies are stimulated with 627 nm LEDs. $n = 20$–39 flies per genotype. The raw data underlying parts A, B, and C can be found in S1 Data.
(TIF)

**S9 Fig. Vehicle-fed sleep data for males in optogenetic experiments. (A)** Box plots of locomotor activity counts per minute awake for vehicle-fed flies presented in S6A Fig. Two-way repeated measures ANOVA followed by Sidak's multiple comparisons test found no difference

in locomotor activity per awake time when the flies are stimulated with 627 nm LEDs. $n$ = 20–40 flies per genotype. **(B)** Box plots of daytime sleep bout duration for vehicle-fed flies presented in S6A Fig. Two-way repeated measures ANOVA followed by Sidak's multiple comparisons test found that most vehicle-fed sleep-promoting lines show no difference in daytime sleep bout duration when the flies are stimulated with 627 nm LEDs. \*$P < 0.05$, \*\*$P < 0.01$, \*\*\*\*$P < 0.0001$. $n$ = 20–40 flies per genotype. **(C)** Box plots of nighttime sleep bout duration for vehicle-fed flies presented in S6A Fig. Two-way repeated measures ANOVA followed by Sidak's multiple comparisons test show no difference in nighttime sleep bout duration when vehicle-fed flies are stimulated with 627 nm LEDs. $n$ = 20–40 flies per genotype. The raw data underlying parts A, B, and C can be found in S1 Data.
(TIF)

**S10 Fig. VNC-SP neurons are present in sleep-promoting FBS lines. (A)** Representative confocal stacks of *FBS42>UAS-GFP* (left) and *FBS42>UAS-GFP; ChAT-LexA>2x LexAop-2KZip⁺* (right) female flies showing the location of the VNC-SP bowtie processes, the dFB region, the VNC and VNC-SP neurons. Expression of GFP in VNC-SP neurons and in their bowtie brain processes is abolished by the expression of the KZip⁺ repressor. Gray arrows show VNC-SP neurons. Green, anti-GFP; magenta, anti-nc82. **(B)** Representative confocal stacks of *FBS45>UAS-GFP* (left) and *FBS45>UAS-GFP; ChAT-LexA> LexAop2KZip⁺* (right) female flies showing the location of the VNC-SP bowtie processes, the dFB region, the VNC and VNC-SP neurons. Expression of GFP in VNC-SP neurons and in their bowtie brain processes is abolished by the expression of the KZip⁺ repressor. Gray arrows show VNC-SP neurons. Green, anti-GFP; magenta, anti-nc82. **(C)** Representative confocal stacks of *FBS53>UAS-GFP* (left) and *FBS53>UAS-GFP; ChAT-LexA> LexAop2KZip⁺* (right) female flies showing the location of the VNC-SP bowtie processes, the dFB region, the VNC and VNC-SP neurons. Expression of GFP in VNC-SP neurons and in their bowtie brain processes is abolished by the expression of the KZip⁺ repressor. Gray arrows show VNC-SP neurons. Green, anti-GFP; magenta, anti-nc82. **(D)** Representative confocal stacks of *FBS68>UAS-GFP* (left) and *FBS68>UAS-GFP; ChAT-LexA> 2x LexAop2KZip⁺* (right) female flies showing the location of the VNC-SP bowtie processes, the dFB region, the VNC and VNC-SP neurons. Expression of GFP in VNC-SP neurons and in their bowtie brain processes is abolished by the expression of the KZip⁺ repressor. Gray arrows show VNC-SP neurons. Green, anti-GFP; magenta, anti-nc82. **(E)** Box plots of total sleep change in % ((total sleep on day 3-total sleep on day 2/total sleep on day 2) × 100) for female control (Empty: Empty-AD; 23E10-DBD) and 4 sleep-promoting FBS lines expressing UAS-TrpA1 or UAS-TrpA1 and the KZip⁺ repressor in cholinergic neurons (ChAT-KZip⁺). Two-way ANOVA followed by Sidak's multiple comparisons revealed that repressing the expression of TrpA1 in cholinergic neurons abolishes the sleep increases obtained when thermogenetically activating all 4 FBS lines. \*\*\*\*$P < 0.0001$, n.s. = not significant, # indicates that sleep is significantly increased compared with *Empty>TrpA1* controls. $n$ = 17–69 flies per genotype. The raw data underlying part E can be found in S1 Data.
(TIF)

**S11 Fig. Differential strength of expression in VNC-SP neurons for different FBS lines. (A)** Schematic of the GFP.DD experimental design. **(B)** Representative confocal stacks of females 23E10-GAL4, VNC-SP Split, FBS1, FBS33, FBS45, and FBS58 expressing GFP-DD and fed DMSO or TMP for 24 h before dissection. Stacks are focused on the metathoracic region of the VNC where VNC-SP cell bodies are located. Green, anti-GFP. White asterisks indicate VNC-SP cell bodies. **(C)** Quantification of staining intensity for data presented in B. Two-way ANOVA followed by Sidak's multiple comparisons revealed that there is

significantly more GFP staining in VNC-SP neurons in 23E10-GAL4, VNC-SP Split, FBS33, and FBS45 fed TMP compared with DMSO fed flies. n.s = not significant, **$P < 0.01$, ****$P < 0.0001$. $n = 5$–14 VNC per genotype and condition. The raw data underlying part C can be found in S1 Data.
(TIF)

**S12 Fig. Expression pattern of dFB-Split. Representative confocal stack images of adult tissues from *Empty-AD; 23E10-DBD> UAS-mCD8GFP* (top) and *dFB-Split> UAS-mCD8GFP* (bottom) female flies.** GFP is expressed in the brain, but not the wings, legs, ovaries, or gut in dFB-Split. Tissue was dissected, fixed, and stained with DAPI. Green, anti-GFP; Blue, DAPI.
(TIF)

**S13 Fig. Additional data for optogenetic activation of dFB$^{23E10 \cap 84C10}$ neurons at 10 Hz. (A)** Box plots of locomotor activity counts per minute awake for retinal-fed flies presented in Fig 1G. The bottom and top of each box represents the first and third quartile, and the horizontal line dividing the box is the median. The whiskers represent the 10th and 90th percentiles. Two-way repeated measures ANOVA followed by Sidak's multiple comparisons test found no difference in locomotor activity per awake time when the flies are stimulated at 10 Hz with 627 nm LEDs. n.s = not significant. $n = 23$–24 flies per genotype. **(B)** Box plots of daytime sleep bout duration (in minutes) for retinal-fed flies presented in Fig 1G. The bottom and top of each box represents the first and third quartile, and the horizontal line dividing the box is the median. The whiskers represent the 10th and 90th percentiles. Two-way repeated measures ANOVA followed by Sidak's multiple comparisons indicate that daytime sleep bout duration is increased in *dFB-Split>CsChrimson* female flies when stimulated with 627 nm LEDs at 10 Hz. n.s = not significant, ***$P < 0.001$. $n = 23$–24 flies per genotype. **(C)** Box plots of nighttime sleep bout duration (in minutes) for retinal-fed flies presented in Fig 1G. The bottom and top of each box represents the first and third quartile, and the horizontal line dividing the box is the median. The whiskers represent the 10th and 90th percentiles. Two-way repeated measures ANOVA followed by Sidak's multiple comparisons found no differences in nighttime sleep bout duration. n.s = not significant. $n = 23$–24 flies per genotype. **(D)** Box plots of locomotor activity counts per minute awake for vehicle-fed flies presented in Fig 1G. The bottom and top of each box represents the first and third quartile, and the horizontal line dividing the box is the median. The whiskers represent the 10th and 90th percentiles. Two-way repeated measures ANOVA followed by Sidak's multiple comparisons test found no difference in locomotor activity per awake time when the flies are stimulated at 10 Hz with 627 nm LEDs. n.s = not significant. $n = 22$ flies per genotype. **(E)** Box plots of daytime sleep bout duration (in minutes) for vehicle-fed flies presented in Fig 1G. The bottom and top of each box represents the first and third quartile, and the horizontal line dividing the box is the median. The whiskers represent the 10th and 90th percentiles. Two-way repeated measures ANOVA followed by Sidak's multiple comparisons found no differences in daytime sleep bout duration. n.s = not significant. $n = 22$ flies per genotype. **(F)** Box plots of nighttime sleep bout duration (in minutes) for vehicle-fed flies presented in Fig 1G. The bottom and top of each box represents the first and third quartile, and the horizontal line dividing the box is the median. The whiskers represent the 10th and 90th percentiles. Two-way repeated measures ANOVA followed by Sidak's multiple comparisons. ****$P < 0.0001$. $n = 22$ flies per genotype. The raw data underlying parts A, B, C, D, E, and F can be found in S1 Data.
(TIF)

**S14 Fig. Additional data for optogenetic activation of dFB$^{23E10 \cap 84C10}$ neurons at 20 Hz. (A)** Box plots of locomotor activity counts per minute awake for retinal-fed flies presented in Fig 1I. Two-way repeated measures ANOVA followed by Sidak's multiple comparisons test found

no difference in locomotor activity per awake time when the flies are stimulated with 627 nm LEDs. n.s. = not significant. *n* = 34–38 flies per genotype. **(B)** Box plots of daytime sleep bout duration for retinal-fed flies presented in Fig 1I. Two-way repeated measures ANOVA followed by Sidak's multiple comparisons test found that daytime sleep bout duration is significantly increased in *dFB-Split>CsChrimson* female flies that are stimulated with 627 nm LEDs. ****$P < 0.0001$, n.s. = not significant. *n* = 34–38 flies per genotype. **(C)** Box plots of nighttime sleep bout duration for retinal-fed flies presented in Fig 1I. Two-way repeated measures ANOVA followed by Sidak's multiple comparisons test found that nighttime sleep bout duration is significantly increased in *dFB-Split>CsChrimson* female flies that are stimulated with 627 nm LEDs. *$P < 0.05$, n.s. = not significant. *n* = 34–38 flies per genotype. **(D)** Box plots of locomotor activity counts per minute awake for vehicle-fed flies presented in Fig 1I. Two-way repeated measures ANOVA followed by Sidak's multiple comparisons test found no difference in locomotor activity per awake time when the flies are stimulated with 627 nm LEDs. n.s. = not significant. *n* = 26–32 flies per genotype. **(E)** Box plots of daytime sleep bout duration for vehicle-fed flies presented in Fig 1I. Two-way repeated measures ANOVA followed by Sidak's multiple comparisons test found no difference in daytime sleep bout duration when the flies are stimulated with 627 nm LEDs. n.s. = not significant. *n* = 26–32 flies per genotype. **(F)** Box plots of nighttime sleep bout duration for vehicle-fed flies presented in Fig 1I. Two-way repeated measures ANOVA followed by Sidak's multiple comparisons test. ****$P < 0.0001$. *n* = 26–32 flies per genotype. **(G)** Box plots of total sleep change in % for male control (Empty-AD; 23E10-DBD) and dFB-Split flies expressing CsChrimson under 20 Hz optogenetic activation. Two-way ANOVA followed by Sidak's multiple comparisons revealed that sleep is significantly increased in *dFB-Split>CsChrimson* males. n.s. = not significant, *$P < 0.05$. *n* = 31–41 flies per genotype and condition. **(H)** Box plots of locomotor activity counts per minute awake for retinal-fed flies presented in G. Two-way repeated measures ANOVA followed by Sidak's multiple comparisons test found no difference in locomotor activity per awake time when the flies are stimulated with 627 nm LEDs. n.s. = not significant. *n* = 35–41 flies per genotype. **(I)** Box plots of daytime sleep bout duration for retinal-fed flies presented in G. Two-way repeated measures ANOVA followed by Sidak's multiple comparisons test found that daytime sleep bout duration is significantly increased in *dFB-Split>CsChrimson* male flies that are stimulated with 627 nm LEDs. ****$P < 0.0001$, n.s. = not significant. *n* = 35–41 flies per genotype. **(J)** Box plots of nighttime sleep bout duration for retinal-fed flies presented in G. Two-way repeated measures ANOVA followed by Sidak's multiple comparisons test found that nighttime sleep bout duration is significantly increased in *dFB-Split>CsChrimson* male flies that are stimulated with 627 nm LEDs. ****$P < 0.0001$, n.s. = not significant. *n* = 35–41 flies per genotype. **(K)** Box plots of locomotor activity counts per minute awake for vehicle-fed flies presented in G. Two-way repeated measures ANOVA followed by Sidak's multiple comparisons test. ***$P < 0.001$, ****$P < 0.0001$. *n* = 31–36 flies per genotype. **(L)** Box plots of daytime sleep bout duration for vehicle-fed flies presented in G. Two-way repeated measures ANOVA followed by Sidak's multiple comparisons test found no difference in vehicle-fed males. n.s. = not significant. *n* = 31–36 flies per genotype. **(M)** Box plots of nighttime sleep bout duration for vehicle-fed flies presented in G. Two-way repeated measures ANOVA followed by Sidak's multiple comparisons test show no difference in nighttime sleep bout duration when vehicle-fed flies are stimulated with 627 nm LEDs. n.s. = not significant. *n* = 31–36 flies per genotype. The raw data underlying parts A, B, C, D, E, F, G, H, I, J, K, L, and M can be found in S1 Data.
(TIF)

**S15 Fig. Additional data for optogenetic activation of dFB[23E10∩84C10] neurons at 50 Hz. (A)** Box plots of locomotor activity counts per minute awake for retinal-fed flies presented in

Fig 1K. Two-way repeated measures ANOVA followed by Sidak's multiple comparisons test found no difference in locomotor activity per awake time in *dFB-Split>CsChrimson* female flies that are stimulated with 627 nm LEDs. \*\*$P < 0.01$, n.s. = not significant. $n = 33–38$ flies per genotype. **(B)** Box plots of daytime sleep bout duration for retinal-fed flies presented in Fig 1K. Two-way repeated measures ANOVA followed by Sidak's multiple comparisons test found that daytime sleep bout duration is significantly increased in *dFB-Split>CsChrimson* female flies that are stimulated with 627 nm LEDs. \*\*\*\*$P < 0.0001$, n.s. = not significant. $n = 33–38$ flies per genotype. **(C)** Box plots of nighttime sleep bout duration for retinal-fed flies presented in Fig 1K. Two-way repeated measures ANOVA followed by Sidak's multiple comparisons test found that nighttime sleep bout duration is significantly increased in *dFB-Split>CsChrimson* female flies that are stimulated with 627 nm LEDs. \*\*\*\*$P < 0.0001$, n.s. = not significant. $n = 33–38$ flies per genotype. **(D)** Box plots of locomotor activity counts per minute awake for vehicle-fed flies presented in Fig 1K. Two-way repeated measures ANOVA followed by Sidak's multiple comparisons test. \*$P < 0.05$, n.s. = not significant. $n = 25–31$ flies per genotype. **(E)** Box plots of daytime sleep bout duration for vehicle-fed flies presented in Fig 1K. Two-way repeated measures ANOVA followed by Sidak's multiple comparisons test found no difference in daytime sleep bout duration when the flies are stimulated with 627 nm LEDs. n.s. = not significant. $n = 25–31$ flies per genotype. **(F)** Box plots of nighttime sleep bout duration for vehicle-fed flies presented in Fig 1K. Two-way repeated measures ANOVA followed by Sidak's multiple comparisons test. \*\*$P < 0.01$, n.s. = not significant. $n = 25–31$ flies per genotype. **(G)** Box plots of total sleep change in % for male control (Empty-AD; 23E10-DBD) and dFB-Split flies expressing CsChrimson under 50 Hz optogenetic activation. Two-way ANOVA followed by Sidak's multiple comparisons revealed that sleep is significantly increased in *dFB-Split>CsChrimson* males. n.s. = not significant, \*\*\*$P < 0.001$. $n = 62–73$ flies per genotype and condition. **(H)** Box plots of locomotor activity counts per minute awake for retinal-fed flies presented in G. Two-way repeated measures ANOVA followed by Sidak's multiple comparisons test. \*\*\*\*$P < 0.0001$, n.s. = not significant. $n = 62–71$ flies per genotype. **(I)** Box plots of daytime sleep bout duration for retinal-fed flies presented in G. Two-way repeated measures ANOVA followed by Sidak's multiple comparisons test found that daytime sleep bout duration is significantly increased in *dFB-Split>CsChrimson* female flies that are stimulated with 627 nm LEDs. \*\*\*\*$P < 0.0001$, n.s. = not significant. $n = 62–71$ flies per genotype. **(J)** Box plots of nighttime sleep bout duration for retinal-fed flies presented in G. Two-way repeated measures ANOVA followed by Sidak's multiple comparisons test show no difference in nighttime sleep bout duration when retinal-fed flies are stimulated with 627 nm LEDs. n.s. = not significant. $n = 62–71$ flies per genotype. **(K)** Box plots of locomotor activity counts per minute awake for vehicle-fed flies presented in G. Two-way repeated measures ANOVA followed by Sidak's multiple comparisons test. \*\*$P < 0.01$, n.s. = not significant. $n = 63–73$ flies per genotype. **(L)** Box plots of daytime sleep bout duration for vehicle-fed flies presented in G. Two-way repeated measures ANOVA followed by Sidak's multiple comparisons test found no difference in daytime sleep bout duration when the flies are stimulated with 627 nm LEDs. n.s. = not significant. $n = 63–73$ flies per genotype. **(M)** Box plots of nighttime sleep bout duration for vehicle-fed flies presented in G. Two-way repeated measures ANOVA followed by Sidak's multiple comparisons test. \*\*$P < 0.01$. $n = 63–73$ flies per genotype. The raw data underlying parts A, B, C, D, E, F, G, H, I, J, K, L, and M can be found in S1 Data. (TIF)

**S16 Fig. Validation of the multibeam system. (A)** Setup for combined multibeam and video analysis. **(B)** Observed % time spent in each of 7 behaviors (Rest, Walking, Feeding, Grooming, Posture change, Proboscis extension, and single leg movement) for 19 *Canton-S* flies during 10 min recording. **(C)** Sensitivity (ratio of minutes with a behavior identified by the

multibeam system to all visually labeled minutes of the same behavior) and accuracy (ratio of correctly identified minutes with a behavior by the multibeam system to all identified minutes with the same behavior by the multibeam system) of the multibeam system for identifying all behaviors. For the accuracy calculation, all micromovements (Feeding, Grooming, Posture change, Proboscis extension, and single leg movement) were pooled together. **(D)** Comparison of multibeam analysis and video observation for 4 *Canton-S* flies during 10 min. **(E)** Comparison of multibeam analysis and video observation for the 4 *Canton-S* flies shown in Movie 1. The raw data underlying parts B, C, and E can be found in S1 Data.
(TIF)

**S17 Fig. Courtship indices for untrained and trained flies. (A)** Courtship index values for male flies in untrained and trained groups for each condition presented in Fig 2C. Unpaired parametric *t* test for each group of untrained and trained males for each condition. Sample size (untrained:trained) from left to right on the graph, $n = 54$ (27:27), 51 (27:24), 49 (27:22), 51 (26:25), 46 (21:25), and 53 (27:26), respectively. Courtship indices <10% were excluded from both groups. **$P < 0.01$, ns = not significant. The raw data underlying part A can be found in S1 Data.
(TIF)

**S18 Fig. Additional data for hyperpolarization of dFB$^{23E10 \cap 84C10}$ neurons. (A)** Representative brain confocal stack of *dFB-Split>UAS-mCD8GFP*. Green, anti-GFP; magenta, anti-nc82. **(B)** Representative brain confocal stack of *dFB-Split>UAS-Kir2.1.EGFP*. We observed $23.40 \pm 0.75$ ($n = 5$) dFB$^{23E10 \cap 84C10}$ neurons in *dFB-Split>UAS-Kir2.1.EGFP* brains. This number is similar to what we have observed for *dFB-Split>UAS-mCD8GFP* brains (Fig 1A). Green, anti-GFP; magenta, anti-nc82. **(C)** Box plots of total sleep (in minutes) for control and *dFB-Split>Kir2.1* male flies. A two-tailed Mann–Whitney U test revealed that *dFB-Split>Kir2.1* male flies sleep significantly more than controls. ****$P < 0.0001$. $n = 69$–75 flies per genotype. **(D)** Box plots of locomotor activity counts per minute awake for flies presented in C. A two-tailed Mann–Whitney U test revealed no differences between controls and *dFB-Split>Kir2.1* male flies. n.s. = not significant. $n = 69$–75 flies per genotype. **(E)** Box plots of daytime sleep bout duration (in minutes) for flies presented in C. A two-tailed Mann–Whitney U test revealed that daytime sleep bout duration is increased in *dFB-Split>Kir2.1* male flies. **$P < 0.01$. $n = 69$–75 flies per genotype. **(F)** Box plots of nighttime sleep bout duration (in minutes) for flies presented in C. A two-tailed Mann–Whitney U test revealed that nighttime sleep bout duration is increased in *dFB-Split>Kir2.1* male flies. ***$P < 0.001$. $n = 69$–75 flies per genotype. The raw data underlying parts B, C, D, E, and F can be found in S1 Data.
(TIF)

**S19 Fig. The dFB is not GABAergic. (A)** Representative confocal stack of a female *23E10-GAL4>UAS-mCD8GFP* brain stained with GFP and GABA antibodies and focusing on dFB cell bodies. Green, anti-GFP; magenta, anti-GABA. **(B)** Representative confocal stack of a female *Gad1-AD; 23E10-DBD>UAS-mCD8GFP* brain and VNC. Green, anti-GFP; magenta, anti-nc82 (neuropile marker). **(C)** Representative confocal stack of a female *VGlut-AD (84713); 23E10-DBD>UAS-mCD8GFP* brain. Yellow and red arrows show non-dFB neurons. Green, anti-GFP; magenta, anti-nc82 (neuropile marker). **(D)** Representative confocal stack of a female *VGlut-AD (82986); 23E10-DBD>UAS-mCD8GFP* brain. Yellow and red arrows show non-dFB neurons. Green, anti-GFP; magenta, anti-nc82 (neuropile marker).
(TIF)

**S20 Fig. Additional data for optogenetic activation of dFB$^{VGlut \cap 84C10}$ and dFB$^{ChAT \cap 84C10}$ neurons. (A)** Box plots of total sleep change in % for female control (Empty-Split), VGlut-AD;

84C10-DBD, and 84C10-AD; ChAT-DBD flies expressing CsChrimson under regular opto-genetic activation (5 ms LED ON, 95 ms LED OFF, with a 4 s delay between pulses). Two-way ANOVA followed by Sidak's multiple comparisons found no difference. n.s. = not significant. $n$ = 12–36 flies per genotype and condition. **(B)** Box plots of total sleep change in % for female control (Empty-Split), VGlut-AD; 84C10-DBD, and 84C10-AD; ChAT-DBD flies expressing CsChrimson under 20 Hz activation. Two-way ANOVA followed by Sidak's multiple comparisons found that total sleep is increased in *84C10-AD; ChAT-DBD>CsChrimson*. n.s. = not significant, ****$P < 0.0001$. $n$ = 19–26 flies per genotype and condition. **(C)** Box plots of nighttime sleep change in % ((nighttime sleep on activation day-nighttime sleep on baseline day/nighttime sleep on baseline day) × 100) for control (*Empty-Split>CsChrimson*), *VGlut-AD; 84C10-DBD>CsChrimson* and *84C10-AD; ChAT-DBD>CsChrimson* female flies presented in Fig 4H–K. Two-way ANOVA followed by Sidak's multiple comparisons revealed that activating 84C10-AD; ChAT-DBD neurons significantly increases nighttime sleep. n.s. = not significant, ****$P < 0.0001$. $n$ = 32–47 flies per genotype and condition. **(D)** Box plots of daytime sleep bout numbers for retinal-fed flies presented in Fig 4K. Two-way repeated measures ANOVA followed by Sidak's multiple comparisons test found that daytime sleep bout numbers are significantly increased in *VGlut-AD; 84C10-DBD>CsChrimson* and significantly decreased in *84C10-AD; ChAT-DBD>CsChrimson* female flies that are stimulated with 627 nm LEDs. n.s. = not significant, **$P < 0.01$, ****$P < 0.0001$. $n$ = 33–47 flies per genotype. **(E)** Box plots of daytime sleep bout duration for retinal-fed flies presented in Fig 4K. Two-way repeated measures ANOVA followed by Sidak's multiple comparisons test found that daytime sleep bout duration is significantly increased in *84C10-AD; ChAT-DBD>CsChrimson* female flies. n.s. = not significant, ****$P < 0.0001$. $n$ = 33–47 flies per genotype. **(F)** Box plots of locomotor activity counts per minute awake for retinal-fed flies presented in Fig 4K. Two-way repeated measures ANOVA followed by Sidak's multiple comparisons revealed that activating 84C10-AD; ChAT-DBD neurons significantly increases locomotion when the flies are awake. n.s. = not significant, **$P < 0.01$, ****$P < 0.0001$. $n$ = 33–47 flies per genotype. **(G)** Box plots of nighttime sleep bout numbers for retinal-fed flies presented in Fig 4K. Two-way repeated measures ANOVA followed by Sidak's multiple comparisons test found that night-time sleep bout numbers are significantly increased in *VGlut-AD; 84C10-DBD>CsChrimson* and significantly decreased in *84C10-AD; ChAT-DBD>CsChrimson* female flies that are stimulated with 627 nm LEDs. n.s. = not significant, **$P < 0.01$, ****$P < 0.0001$. $n$ = 33–47 flies per genotype. **(H)** Box plots of nighttime sleep bout duration for retinal-fed flies presented in Fig 4K. Two-way repeated measures ANOVA followed by Sidak's multiple comparisons test found that nighttime sleep bout duration is significantly increased in *84C10-AD; ChAT-DBD>CsChrimson* female flies. n.s. = not significant, *$P < 0.05$, ****$P < 0.0001$. $n$ = 33–47 flies per genotype. **(I)** Box plots of daytime sleep bout numbers for vehicle-fed flies presented in Fig 4K. Two-way repeated measures ANOVA followed by Sidak's multiple comparisons test. n.s. = not significant, *$P < 0.05$. $n$ = 32–45 flies per genotype. **(J)** Box plots of daytime sleep bout duration for vehicle-fed flies presented in Fig 4K. Two-way repeated measures ANOVA followed by Sidak's multiple comparisons test. n.s. = not significant. $n$ = 32–45 flies per genotype. **(K)** Box plots of locomotor activity counts per minute awake for vehicle-fed flies presented in Fig 4K. Two-way repeated measures ANOVA followed by Sidak's multiple comparisons test. n.s. = not significant, *$P < 0.05$. $n$ = 32–45 flies per genotype. **(L)** Box plots of nighttime sleep bout numbers for vehicle-fed flies presented in Fig 4K. Two-way repeated measures ANOVA followed by Sidak's multiple comparisons test. n.s. = not significant, ****$P < 0.0001$. $n$ = 32–45 flies per genotype. **(M)** Box plots of nighttime sleep bout duration for vehicle-fed flies presented in Fig 4K. Two-way repeated measures ANOVA followed by Sidak's multiple comparisons test. n.s. = not significant, ****$P < 0.0001$. $n$ = 32–45 flies per

genotype. The raw data underlying parts A, B, C, D, E, F, G, H, I, J, K, L, and M can be found in S1 Data.
(TIF)

**S21 Fig. Multibeam data for optogenetic activation of dFB$^{VGlut\cap84C10}$ and dFB$^{ChAT\cap84C10}$ neurons at 50 Hz. (A)** Sleep profile in minutes of sleep per hour for day 2 (LED OFF, blue line) and day 3 (LED ON, red line) for retinal-fed Empty-Split control females expressing CsChrimson subjected to a 50 Hz optogenetic activation protocol (cycles of 5 ms LED ON, 15 ms LED OFF) obtained with the DAM5H multibeam system. **(B)** Sleep profile in minutes of sleep per hour for day 2 (LED OFF, blue line) and day 3 (LED ON, red line) for retinal-fed *VGlut-AD; 84C10-DBD>CsChrimson* female flies subjected to a 50 Hz optogenetic activation protocol (cycles of 5 ms LED ON, 15 ms LED OFF) obtained with the DAM5H multibeam system. **(C)** Sleep profile in minutes of sleep per hour for day 2 (LED OFF, blue line) and day 3 (LED ON, red line) for retinal-fed *84C10-AD; ChAT-DBD>CsChrimson* female flies subjected to a 50 Hz optogenetic activation protocol (cycles of 5 ms LED ON, 15 ms LED OFF) obtained with the DAM5H multibeam system. **(D)** Box plots of total sleep change in % ((total sleep on activation day-total sleep on baseline day/total sleep on baseline day) × 100) obtained with the DAM5H system for control (*Empty-Split>CsChrimson*), *VGlut-AD; 84C10-DBD>CsChrimson* and *84C10-AD; ChAT-DBD>CsChrimson* female flies under a 50 Hz optogenetic activation protocol. Two-way ANOVA followed by Sidak's multiple comparisons revealed that activating 84C10-AD; ChAT-DBD neurons significantly increases total sleep. n.s. = not significant, ****$P < 0.0001$. $n = 19$–$29$ flies per genotype and condition. The raw data underlying part D can be found in S1 Data.
(TIF)

**S1 Table. Description of FBS lines.** Identification of the AD construct combined with 23E10-DBD in each FBS line. Average number ± SEM and range of dFB neurons labeled by each line. Bowtie-VNC-SP (Y/N) indicates whether "bowtie" processes are seen in the brain and VNC-SP cell bodies are present in the VNC, Y = yes, N = no. Range of additional (non-dFB) neurons labeled in the brain of each line. Average number ± SEM of VNC metathoracic cells labeled by each FBS line. TPN1 (Y/N) indicates whether TPN1 neurons are present in the expression pattern, Y = yes, N = no. Range of additional cells in the VNC include any VNC neurons that is not TPN1 or VNC-SP. The raw data underlying S1 Table can be found in S1 Data.
(DOCX)

**S2 Table. Summary of sleep phenotypes obtained for thermogenetic and optogenetic activation of FBS lines.** X indicates a significant increase between baseline and activation days, (+) indicates a significant increase compared with controls during thermogenetic activation, and (−) indicates a significant decrease.
(DOCX)

**S3 Table. List of fly stocks used in this study.**
(DOCX)

**S1 Movie. Brain of a dFB-Split>UAS-GFP female.**
(AVI)

**S2 Movie. VNC of a dFB-Split>UAS-GFP female.**
(AVI)

**S3 Movie. Brain of a 84C10AD; ChAT-DBD>UAS-GFP female.**
(AVI)

**S4 Movie. VNC of a 84C10AD; ChAT-DBD>UAS-GFP female.**
(AVI)

**S5 Movie. Brain of a VGlut-AD; 84C10-DBD>UAS-GFP female.**
(AVI)

**S6 Movie. VNC of a VGlut-AD; 84C10-DBD>UAS-GFP female.**
(AVI)

**S7 Movie. Brain of a VGlut-AD (84713); 23E10-DBD>UAS-GFP female.**
(AVI)

**S8 Movie. Brain of a VGlut-AD (82986); 23E10-DBD>UAS-GFP female.**
(AVI)

**S1 Data. Raw data underlying all Figs.**
(XLSX)

## Acknowledgments

We thank Aaron DiAntonio, Kausik Si, Gerry Rubin, and Paul Shaw for sharing reagents and protocols.

## Author contributions

**Conceptualization:** Stephane Dissel.

**Data curation:** Joseph D. Jones, Brandon L. Holder, Andrew C. Montgomery, Chloe V. McAdams, Emily He, Anna E. Burns, Kiran R. Eiken, Alex Vogt, Adriana I. Velarde, Alexandra J. Elder, Jennifer A. McEllin, Stephane Dissel.

**Funding acquisition:** Stephane Dissel.

**Investigation:** Joseph D. Jones, Brandon L. Holder, Andrew C. Montgomery, Chloe V. McAdams, Emily He, Anna E. Burns, Kiran R. Eiken, Alex Vogt, Adriana I. Velarde, Alexandra J. Elder, Jennifer A. McEllin, Stephane Dissel.

**Methodology:** Joseph D. Jones, Brandon L. Holder, Andrew C. Montgomery, Jennifer A. McEllin, Stephane Dissel.

**Project administration:** Stephane Dissel.

**Supervision:** Stephane Dissel.

**Visualization:** Joseph D. Jones, Brandon L. Holder, Andrew C. Montgomery, Jennifer A. McEllin, Stephane Dissel.

**Writing – original draft:** Joseph D. Jones, Brandon L. Holder, Andrew C. Montgomery, Jennifer A. McEllin, Stephane Dissel.

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
