## [Editor Report · Decision Letter 0]

17 Apr 2024

Dear Dr Dissel, 

Thank you for submitting your manuscript entitled "The dorsal fan-shaped body is a neurochemically heterogeneous sleep-regulating center in Drosophila" for consideration as a Research Article by PLOS Biology.

Your manuscript has now been evaluated by the PLOS Biology editorial staff as well as by an academic editor with relevant expertise and I am writing to let you know that we would like to send your submission out for external peer review as an *Update Article*.

Once your full submission is complete, your paper will undergo a series of checks in preparation for peer review. After your manuscript has passed the checks it will be sent out for review. To provide the metadata for your submission, please Login to Editorial Manager (https://www.editorialmanager.com/pbiology) within two working days, i.e. by Apr 19 2024 11:59PM.

Kind regards,

Christian

Christian Schnell, PhD

Senior Editor

PLOS Biology

cschnell@plos.org

---

## [Decision Letter · Decision Letter 1]

25 May 2024

Dear Dr Dissel,

Thank you for your patience while your manuscript "The dorsal fan-shaped body is a neurochemically heterogeneous sleep-regulating center in Drosophila" was peer-reviewed at PLOS Biology. It has now been evaluated by the PLOS Biology editors, an Academic Editor with relevant expertise, and by several independent reviewers. 

In light of the reviews, which you will find at the end of this email, we would like to invite you to revise the work to thoroughly address the reviewers' reports.

As you will see below, the reviewers find the topic of your study important and interesting, and your study overall well conducted. However, they raise a number of concerns where the claims are not fully supported. We are aware that addressing these will require a large amount of work, but we think that it is critical for the field to resolve the dFB's role in sleep as rigorously as possible to avoid endless debate in the literature.

Given the extent of revision needed, we cannot make a decision about publication until we have seen the revised manuscript and your response to the reviewers' comments. Your revised manuscript is likely to be sent for further evaluation by all or a subset of the reviewers.

**IMPORTANT - SUBMITTING YOUR REVISION**

*Re-submission Checklist*

*Published Peer Review*

*PLOS Data Policy*

*Blot and Gel Data Policy*

Sincerely,

Christian

Christian Schnell, PhD

Senior Editor

PLOS Biology

cschnell@plos.org

REVIEWS:

Reviewer #1: In this manuscript from Stephane Dissel's group, Jones et al present a very thorough investigation of the role of the dorsal fan-shaped body (dFSB) in sleep. This is a very important paper to current investigations of Drosophila sleep, and publication is highly recommended once the concerns outlined below are addressed.

Major revisions:

1. Given that previous studies have proposed that the dFSB is GABAergic, and that the 23E10∩84C10 split does not necessarily label all the dFSB neurons that 84C10 does, the authors should also perform an antibody stain of 23E10-Gal4>UAS-mCD8-GFP, in case the dFSB neurons that are not part of the overlap are GABAergic.

2. To address the fact that constitutive KIR2.1 expression in the 23E10∩84C10 neurons increases sleep, the authors should use a temperature-sensitive repressor (tubGal80ts) to silence the neurons during adulthood. Shibire is only effective at silencing chemical synapses, and Troup et al (eLife 2018) showed that the dFSB has gap-junction coupling (INX6); thus, using the temperature-sensitive repressor on KIR2.1 will address the possibility that loss of the gap junction coupling is necessary for sleep. 

3. An alternative explanation for why constitutive KIR2.1 silencing increases sleep may be because of the imbalance created by silencing a population of heterogeneous neurons. To address this, the authors should test how using the cholinergic 84C10 split-Gal4 and glutamatergic 84C10 split-Gal4 alter sleep when used to drive KIR2.1. Only the csChrimson results of these drivers are shown.

Minor revisions:

1. In both the abstract (L44-46) and the introduction (L120-122), the authors refer to the recent De et al 2023 Current Biology paper, stating that De et al's conclusions were based on genetic tools that are not dFB-specific. While De et al do use 23E10-Gal4 to test homeostatic sleep, a more problematic issue with that study, leading to a discrepancy between their results and the ones presented by Jones et al in 2021 for an identical genetic manipulation, is their use of a very mild deprivation protocol. Mentioning the mild deprivation protocol is important for resolving an existing controversy in the literature.

2. Related to point #1, a major conclusion of this manuscript seems to be that the dFSB requires stronger stimulation in order to promote sleep than driver lines that includes VNC neurons, and this should be mentioned in the abstract.

3. In L164-165, the authors state, "This increase is likely explained by the need for a fly to perform tasks that are mutually exclusive to sleep in a reduced amount of waking time." - This sentence could be argued to be anthropomorphizing, and a more reasonable interpretation may be that a fly experiencing more restorative sleep has more energy for higher activity.

4. In Table S2, what is the difference between "X", "(-)", and "(+)"? 

5. Include supplementary video comparing the flies doing different behaviors with and without csChrimson activation. Given the current concern in the field about whether neuronal activation causes sleep or some other form of immobility, as well as the trend towards characterizing micromovements and postural adjustments, sharing video documentation would be hugely beneficial to the field.

6. Fig 4C: Less confusing to say "GFP+ only" and "GFP+ total" and "ChAT+GFP" than to say "nothing".

7. Fig 4G: Having a blue square for Gad1-DBD, when there is no bar in the graph, is distracting and confusing.

- 

Reviewer #2: In the Drosophila brain, a group of dorsal fan-shaped body neurons marked by the 23E10-GAL4 driver line was found to be necessary and sufficient for promoting sleep. However, a recent study from the same lab revealed that the 23E10-GAL4 driver line also labels neurons outside of the brain, namely in the ventral nerve cord, which promote sleep when artificially activated. Consequently, behavioral results obtained with 23E10-GAL4 might be inconclusive, as the observed effects could be due to neurons in either the ventral nerve cord or in the dorsal fan-shaped body. To address this issue, the authors of this manuscript have identified driver lines that specifically label neurons in the dorsal fan-shaped body but not in the ventral nerve cord, demonstrating that activation of these neurons promotes sleep. This manuscript thus resolves a significant issue in the field of Drosophila sleep research. I recommend this manuscript for publication, provided that key points are addressed in a revised version.

Key points:

* Figure 3: Here, the empty-split > Kir 2.1 group sleeps abnormally little (especially when compared to the empty-split > CsChrimson). Thus, the sleep-promoting effect observed upon Kir2.1 expression using the dFB-Split line might not be a true effect but rather due to the control group. The authors should add an experiment in which they use proper parental controls, i.e., Kir2.1/+, empty-split/+, dFB-split/+, empty-split > Kir2.1, dFB-split > Kir2.1 in order to validate (or not) their conclusion.

* Figure 4: It is unclear to me as to why the authors use split-GAL4 combinations based on 84C10-AD instead of 23E10-DBD or 23E10-AD given that the initial goal of the paper was to understand cellular heterogeneity in 23E10-GAL4 (as for example written in lines 519-521 in the discussion). The authors should test the following split-GAL4 combinations in sleep experiments and also provide anatomical pictures: 23E10-AD/-DBD + Gad1-AD/-DBD and 23E10-AD/-DBD + ChAT-AD/-DBD and 23E10-AD/-DBD + Vglut-AD/-DBD.

* Figure 4: Do the authors think that the release of Acetylcholine or Glutamate is needed for the observed sleep-promoting effect of VGlut-AD;84C10-DBD or ChAT-AD;84C10-DBD? The authors could knock down the respective vesicular transporters and check whether the sleep-promoting effect is still observed upon optogenetic stimulation.

* In general, I find the nomenclature for the split-GAL4 lines used (e.g., FBS6, VNC-Split, dFB-Split) confusing, non-informative and non-consistent throughout the figures of the paper (e.g. Empty-Split in Figure 3 versus Empty-AD;23E10-DBD in Figure 2, is this the same control line?). It would be helpful to change the names to the actual enhancer nomenclature throughout the paper to avoid any confusion for the reader.

Minor points:

* Figure 1C: please indicate what protocol was used for the optogenetic stimulation experiments, was it the standard protocol the authors refer to in Figure 2B? If so, could it be that some lines tested would actually show an effect given that the standard protocol used in Figure 2B is not effective for most non-bowtie-containing lines?

* Figure 2B: please indicate in grey box what 'Standard' refers to.

* Figure 2M/N: please indicate in figure legend what optogenetic stimulation protocol was used.

* Figure 4B: please provide higher-magnification pictures, it is very hard to see any colocalization.

Reviewer #3: This manuscript by Jones et al. adds to a saga of manuscripts investigating the role of the dFB in sleep regulation. The authors screen a combination of split GAL4 lines and show that the VNC component of 23E10 neurons likely accounts for the sleep phenotype observed under "standard" conditions of activation. The dFB-only component does not show any sleep phenotype under standard conditions, but some sleep phenotype does appear under "non-standard" manipulations (e.g., unusually high optogenetic activation and weirdly enough, in which chronic over-activation leads to the same phenotype as chronic inactivation).

The whole manuscript is rather verbose and would benefit from some synthetic editing. There is a lot of data being presented, mostly useful and relevant but not always so (especially in the supplementary figures). The genetic methodology employed is very careful and commendable. However, the physiological and behavioral analysis is less so and often employs self-made solutions that reinvent the wheel and lack necessary validation (for instance, the video analysis or the analysis of arousal threshold) or description (no code or detailed description of the apparatus is provided, nor validation).

I find the work suggestive but inconclusive, at least for a manuscript that clearly aspires to close the discussion once and for all. I think the authors are going in the right direction but, frankly, looking at the results, I am not reaching the same conclusion that the authors are putting forward. To me, the role of the dFB component of 23E10 appears weaker and weaker given that it is visible only under extreme conditions of manipulation.

Here are some more specific comments:

**Figure 1**: Please find a way to combine the results from 1B and 1C either in the same graph or at least in a way that makes it visually easier to compare the same lines across the two different treatments (e.g., same width of the figure and just one panel on top of the other).

**Supplementary Figures 1-6**: The supplementary figures related to the dataset in Figure 1 are a bit sparse. For instance, the classical sleep pattern is shown only for the thermogenetics dataset (S1) but not for the optogenetics dataset. This is important because almost all your subsequent experiments rely on optogenetics, and it's impossible for the reader to compare the results between Figure 1 and, say, Figure 2 or Figure 4.

**Figure 2B**: What does "Standard" mean? The legend says "5ms LED ON, 95ms LED OFF, with a 4-second delay between pulses," but it's not immediately clear how this compares with the other stimulations used, because both 20Hz and 50Hz are indicated as having 5ms ON and 45ms OFF. I suspect that this is a typo and 50Hz is actually 5ms ON, 15ms OFF. I also assume that the 4-second delay still applies to all three conditions. Perhaps a schematic of the pulse protocol would help.

The actual details of the optogenetics protocol used are very important and need to be clarified. I have a growing suspicion that this strong and prolonged activation simply leads to firing fatigue, and that is why your activation and Kir2.1 inactivation phenotypes are identical, both showing increased sleep. This would also explain why you do not see any sleep phenotype with ShiTS. Whichever the mechanism, the fundamental problem with a phenotype observed only when pushing the system beyond commonly adopted limits is that one starts losing control of what is actually going on. For instance, one could argue that some Chrimson is still expressed in the ventral nerve cord, not enough to be detectable through immunofluorescence or to be activated by standard regimes, but just enough to show a phenotype when the system is pushed beyond the limits. The joiner lab used tsh-GAL80 in their analysis and perhaps is worth considering using that too (and, by the way, it is unfair to say that those are "non-specific tools" - they are as specific as any other intersectional approach).

**Line 362, Figure 2MN**: "the DAM5H multibeam system, ruling out the possibility that we misregistered micromovements as sleep." I am afraid the multibeam system cannot rule out the possibility that you misregistered micromovements as sleep. DAM5H has 15 beams for a tube length of about 65 mm and therefore a resolution of 65/15 = 4.3mm. That is larger than the entire body of the animal so there is no guarantee that micromovements like grooming or seizures will be detected. There are now multiple studies that employ validated video tracking to show that 23E10 or 104y activation result in an increase of micromovements and/or seizures, and this is a critical aspect for a manuscript that aspires to set the record straight. Your video paradigm relies on a poorly described "manual scoring" while much more objective tools are now available diffusely employed in the literature.

**Line 401**: "Acute silencing of dFB23E10Ո84C10 neurons has no effect on sleep [and this] further demonstrates that dFB23E10Ո84C10 neurons regulate sleep." I absolutely have no idea why you came to this conclusion. Silencing these neurons with ShiTS shows absolutely no effect, and yet you take this as confirmation that those neurons regulate sleep? I honestly don't get it.

**Figure 3JK**: This experiment is important but complicated by the way it is conducted. Would you be able to see any effect on homeostasis using transient rather than developmental manipulation? Is there any homeostatic effect at all using TrpA1 with dFB23E10Ո84C10? This experiment is important and deserves more care because the entire literature assumes dFB is modulating homeostasis (the somnostat), but you dedicate to this aspect of the phenotype only this one experiment, which is inherently complicated by the way it's conducted.

---

## [Decision Letter · Decision Letter 2]

22 Nov 2024

Dear Stephane,

Thank you for your patience while we considered your revised manuscript "The dorsal fan-shaped body is a neurochemically heterogeneous sleep-regulating center in Drosophila" for consideration as a Update Article at PLOS Biology. Your revised study has now been evaluated by the PLOS Biology editors, the Academic Editor and the original reviewers. 

In light of the reviews, which you will find at the end of this email, we are pleased to offer you the opportunity to address the remaining points from the reviewers in a revision that we anticipate should not take you very long. We will then assess your revised manuscript and your response to the reviewers' comments with our Academic Editor aiming to avoid further rounds of peer-review, although might need to consult with the reviewers, depending on the nature of the revisions.

**IMPORTANT - SUBMITTING YOUR REVISION**

*Resubmission Checklist*

*Published Peer Review*

*PLOS Data Policy*

*Blot and Gel Data Policy*

Sincerely,

Christian

Christian Schnell, PhD

Senior Editor

PLOS Biology

cschnell@plos.org

REVIEWS:

Reviewer #1: Jones et al (2024) did an excellent job addressing the issues that I raised with the previous version of their manuscript. I have one fairly major (but optional) suggestion, which is whether the organization of the paper would benefit from having Figure 3 (the long term memory data) moved to the end, so that characterization of the effect of silencing using the split-Gal4 (Fig 4 and 5) can follow the description of the effects of activating with the split-Gal4. Other than that, I have minor revision requests listed below.

Minor Revisions:

1. L152: Not sure, but I wonder if the authors meant to say "activator" not "actuator" here? I also feel like it should say, "cation channel" rather than "activator" to maintain a consistent/parallel structure with the TrpA1. Maybe to explicitly spell out what the channels are doing, L150-152 should say, "To activate the cells, each individual Split-GAL4" (not necessary for a field expert but would be beneficial if an undergrad or junior grad student were reading the paper).

2. L200-203: You don't describe the optogenetic approach here - I am assuming this is what you refer to as "our standard optogenetic protocol" in L333-334? From the response to the other reviewer, this seems to be the case, but explicitly mentioning it in the text the first time the optogenetic protocol is used will be helpful.

3. L333-334: "Next, we used our standard optogenetic protocol and again obtained no sleep increases when activating dFB23E10Ո84C10 neurons in females (Fig 2D and 2E)-." Remove the hyphen at the end; specify what is meant by "standard" here (It's shown in the figure but I think it's helpful to also have it explicitly spelled out in the text.)

4. Supplementary Movies 3-18 appeared to be missing from the submission?

5. L382 and Movie S3 and S4 caption: does "Cs", refer to "Canton-s" or "CsChrimson"?

Reviewer #2: I would like to thank the authors for addressing most of my concerns and adding a tremendous amount of new data to the manuscript. The new data presented in Figure S21 are really telling and it if there is a way to add them to the main manuscript that would be great. In my opinion, Figure 1 could be added to the supplementary figures, given that most of the lines tested have VNC-SP expression anyhow. That would make it possible to change Figure S21 to Figure 4. But I will leave it to the authors with respect to the decision they would like to make.

I accept the present manuscript provided that the authors address the minor concerns listed below:

- S5-S18 movies: In the movie legend, no details with respect to the optogenetic stimulation can be found. I suggest to add the optogenetic stimulation protocol in the figure legend. This is important because, for example, in movie S18, I can see the flies are highly active during the stimulation phase. Is this a stimulation condition that is supposed to induce sleep? If so, this would significantly affect the conclusions of the paper and it should be addressed.

- In their response to the reviewers, the authors state the following: 'As correctly stated by the Reviewer, our goal was to identify the contribution (if any) of 23E10-GAL4 dFB neurons in sleep regulation. Unfortunately, as demonstrated in S1 Fig and S1 Table, most 23E10-DBD containing Split-GAL4 FBS lines are not dFB-specific. More importantly, most of them contain the previously described sleep-promoting VNC-SP neurons. In addition, there is no 23E10-AD line available, so we are limited to using 23E10-DBD for creating Split-GAL4 tools.' In fact, there are two 23E10-AD lines available from Bloomington (Bloomington numbers 601444 and 601936). It is unclear to me as to why the authors have not performed these experiments, especially given how central this is for the logical flow of the manuscript. Did the authors test these lines? Are they not working?

- Figure 4R: Why do the authors use a temperature of 29°C as opposed of 31°C for sleep deprivation experiments? 

- Lines 232-235: The authors state that '…, 7 are consistently increasing total sleep and sleep consolidation in females and males using both activation protocols'. In Figure 1, only 4 lines are consistently and significantly increasing total sleep using both activation protocols. Can the authors clarify what evidence their claim is based upon? I recommend to rephrase this sentence more precisely and also adjust the discussion accordingly (lines 728-730).

- Lines 315-316: The authors state '… our activation screens and anatomical analysis identified 16 sleep-promoting FBS lines expressing in diverse numbers of dFB neurons.' However, when looking at figure 1, only 4 lines are consistently and significantly increasing total sleep. I recommend to rephrase more precisely. Also, how can an anatomical screen identify sleep-promoting lines?

- Lines 343-344: The authors state '… indicating that it is effective at activating neurons.' Given that different neurons show different physiological responses due to different biophysical/network properties, a given stimulation protocol that works in one neuronal type doesn't necessarily work in a different neuronal type. This is therefore not an argument that a protocol '.. is effective at activating neurons' in general. The protocols need to be adjusted to the type of neurons tested. I recommend that the authors rephrase this sentence.

- Lines 502-503: The authors state 'As seen in S173A and S173B Fig, we observed no defects in dFB23E10Ո84C10 neuron numbers or processes when expressing Kir2.1, compared with controls'. This is difficult to conclude without proper quantification. I recommend to rephrase this sentence or add a quantification.

- Line 514: The authors state '… without disrupting the anatomical properties of these cells'. I recommend to rephrase this sentence to account for the lack of quantification.

- Lines 660-661: The authors state that 'Thus, these data suggest that … use another neuromodulator to promote sleep.' Another possible interpretation is that the remaining vGLUT transcripts (the knockdown being unlikely to be 100%) is sufficient to transmit downstream information. I recommend to rephrase this sentence unless the authors have some evidence for neuromodulators promoting sleep in these cells.

- Lines 665-667: same comment than for lines 660-661

Typos:

- Line 216: typo, 'S5A Fig' instead of 'S54A Fig'

- Line 218: typo, 'S5B Fig' instead of 'S54B Fig'

- Line 220: typo, 'S5C Fig' instead of 'S54C Fig'

- Similar typos found in lines 221, 224

- Line 488: 'S16A Fig.' instead of 'S164A Fig.'

- Line 574: typo, 'Fig 5E and 5G' instead of 'Fig 54E and 54G'

- Line 574: 'S20 Movies' instead of 'S204 Movies'

Reviewer #3: The authors have clearly taken the reviewers' comments seriously, implementing a significant number of new experiments that enhance the manuscript's value. Notably, they added video analysis, as suggested by this reviewer and reviewer #1, conducted a thorough examination of the neurochemical nature of the cells in the 23E10 dFSB cluster, and incorporated a behavioral readout using STM to LTM consolidation.

The genetic manipulation within this work is of a very high standard, surpassing much of the field. Although there have been substantial improvements in the behavioral analysis, I remain unsure if it achieves the level of "unequivocality" it aims for and claims. This research marks an important advancement in the discussion of whether dFSB regulates sleep; however, it is not yet conclusive, in my opinion. The work holds considerable value, particularly in terms of reagents and quality standards. Nevertheless, I would advise caution in making bold claims, as this is unlikely to be the definitive conclusion. I have noted some weaknesses below that I believe impact the work.

-------

Line 47-50: I suggest placing more emphasis on this point in the abstract to highlight its importance.

Line 60-75: If you need guidance on what content to remove, this paragraph could potentially be omitted. It doesn't contribute significantly and reads more like a review.

Line 131: As you did in the abstract, it's prudent to add a caveat regarding the necessity for much stronger activation of dFSB neurons. Additionally, the current data does not sufficiently support the claim that dFSB neurons are involved "especially in sleep homeostasis." Figures 1, 2, 4, and 5 primarily address baseline sleep, and the only main figure that covers sleep deprivation is Figure 3, which involves Kir2.1.

Line 333: I find the use of the term "standard" conditions problematic in this paper, both here and in other sections. Typically, a "standard" implies a broadly accepted protocol within the field, and there is no universally recognized standard for optogenetics. In this case, readers will not be familiar with your lab's specific "standard optogenetic protocol." It would be more effective to provide the precise quantitative and qualitative details instead.

Figure 2: I find the variability between treatments in this figure puzzling. Even when comparing the traces under "LED off" conditions, there's an unexpected degree of inconsistency. Notably, there are two overlapping sets: one between the blue traces in Standard (D), 20Hz (H), and 50Hz (J); and another between 10Hz (F) and 1Hz (B), which appears very different from the first set. I suspect this variability arises because one set of experiments included D, H, J, while another included B, F, with a single replicate for each. Although the number of flies (N) seems appropriate, it would be beneficial to provide more details on how often each experiment was replicated, especially since this is presented as the result that "unequivocally" resolves the issue.

Line 372 - I think you need a reference if you want to claim that 31C is "more commonly used". TrpA1 isoform B has an activation threshold of 27.8 ± 0.4 °C (Kang and Garrity, Nature 2012 ) so 29C is a commonly used activation temperature in behaviour. In fact, my feeling is that it may be even more commonly used in the sleep field exactly to avoid all those annoying confounding factors that derive from using 31C, as you correctly list at lines L174-178. As you correctly point out at L195 "it may be difficult to fully describe and characterize the sleep behaviors [...] when temperature itself has such a profound effect on sleep".

Line 432 - "we conclude that activation of dFB23E10Ո84C10 neurons increases sleep, rather than micromovements". Please specify the activation conditions. Same for L441.

Line 443: The statement "we conclude that sufficient activation of dFB23E10Ո84C10 neurons can increase sleep" may not be entirely accurate as worded. The use of "sufficient" suggests that there is clear evidence that optogenetic stimulation below 10Hz or TrpA1 is not enough to activate firing, which may not be the case. It would be more appropriate to specify the conditions under which activation leads to increased sleep if such evidence is lacking. Consider revising this statement to accurately reflect the findings and limitations of the study.

L490 - "These results are somewhat surprising as they are similar to the ones obtained when

activating dFB23E10Ո84C10 neurons." You may want to use more qualitative analysis than "similar". Can you run a statistical comparison?

L499 - 527 As I stated in my first round of comments, the Occam's razor possibility to explain this paradoxical effect is that sustained optogenetic activation phenocopies hyperpolarization through synaptic fatigue. There is a physiological limit to how fast and how long a neuron can fire and the fact that they can occasionally fire at 50Hz does not imply that the can sustain that rate for hours in a row (24hr in this case). The old Kandel papers showed that neurons show synaptic fatigue already at 1Hz. At that rate, sooner or later neurons do run out of vescicles and neurotransmitters.

This is admittedly not trivial to test because one would have to measure actual neuronal activity while at the same time stimulating the neuron. It can be done, but it would be too much for me to ask the authors to do this experiment. I would argue, though, that this caveat should be considered in the discussion.

---

## [Decision Letter · Decision Letter 3]

13 Jan 2025

Dear Stephane,

Happy New Year and thank you for your patience while your revised Update Article "The dorsal fan-shaped body is a neurochemically heterogeneous sleep-regulating center in Drosophila" was reviewed by one of the original reviewers and discussed within our team and the Academic Editor for publication in PLOS Biology. On behalf of my colleagues and the Academic Editor, Richard Benton, I am pleased to say that we can in principle accept your manuscript for publication, provided you address any remaining formatting and reporting issues. These will be detailed in an email you should receive within 2-3 business days from our colleagues in the journal operations team; no action is required from you until then. Please note that we will not be able to formally accept your manuscript and schedule it for publication until you have completed any requested changes.

PRESS

Sincerely, 

Christian

Christian Schnell, PhD

Senior Editor

PLOS Biology

cschnell@plos.org